# Dynamic early clusters of nodal proteins contribute to node of Ranvier assembly during myelination of peripheral neurons

Elise LV Malavasi[1], Aniket Ghosh[1], Daniel G Booth[2], Michele Zagnoni[3], Diane L Sherman[1], Peter J Brophy[1]*

[1]Centre for Discovery Brain Sciences, University of Edinburgh, Edinburgh, United Kingdom; [2]Biodiscovery Institute, University of Nottingham, Nottingham, United Kingdom; [3]Centre for Microsystems & Photonics, Dept. Electronic and Electrical Engineering, University of Strathclyde, Strathclyde, United Kingdom

**Abstract** Voltage-gated sodium channels cluster in macromolecular complexes at nodes of Ranvier to promote rapid nerve impulse conduction in vertebrate nerves. Node assembly in peripheral nerves is thought to be initiated at heminodes at the extremities of myelinating Schwann cells, and fusion of heminodes results in the establishment of nodes. Here we show that assembly of 'early clusters' of nodal proteins in the murine axonal membrane precedes heminode formation. The neurofascin (Nfasc) proteins are essential for node assembly, and the formation of early clusters also requires neuronal Nfasc. Early clusters are mobile and their proteins are dynamically recruited by lateral diffusion. They can undergo fusion not only with each other but also with heminodes, thus contributing to the development of nodes in peripheral axons. The formation of early clusters constitutes the earliest stage in peripheral node assembly and expands the repertoire of strategies that have evolved to establish these essential structures.

*For correspondence:
peter.brophy@ed.ac.uk

Competing interests: The authors declare that no competing interests exist.

## Introduction

Rapid propagation of action potentials in the myelinated nerve fibres of vertebrates requires the accumulation of macromolecular complexes containing voltage-gated sodium channels (Nav) at the nodes of Ranvier. Nodes in the peripheral nervous system (PNS) form between axonal segments ensheathed by Schwann cells. Much has been learned about the composition and assembly of nodes, such as the role of proteins encoded by the neurofascin (*Nfasc*) gene, which is essential for node formation; nevertheless, how myelinating Schwann cells initiate the assembly of macromolecular axonal complexes containing Nav is still incompletely understood (*Sherman et al., 2005*; *Zhang et al., 2012*; *Zhang et al., 2020*; *Eshed-Eisenbach et al., 2020*; *Rasband and Peles, 2021*).

Although heminodes are thought to initiate node formation (*Rasband and Peles, 2021*), nodal protein complexes lacking either flanking Schwann cells or myelin on naked peripheral axonal segments have been observed in myelinating co-cultures of Schwann cells with neurons from dorsal root ganglia (DRG) (*Eshed-Eisenbach et al., 2020*), in developing wild-type (WT) mouse sciatic nerves (*Lambert et al., 1997*) and in the peripheral nerves of control mice with floxed alleles that were phenotypically normal (*Eshed-Eisenbach et al., 2020*; *Muir et al., 2014*). In vivo, myelinating axons with these 'node-like clusters' display an enhanced conduction velocity (*Eshed-Eisenbach et al., 2020*), and similar structures accelerate conduction speeds in premyelinated axons of the central nervous system (CNS) (*Freeman et al., 2015*). However, whether 'node-like clusters' in the PNS are simply transient ectopic aggregates or whether there is a mechanistic and/or developmental relationship between their appearance and the ultimate assembly of the mature node of Ranvier remains unclear.

Here we show that 'node-like clusters' are the earliest detectable Nav-rich complexes of nodal proteins in primary sensory neurons, and their appearance requires the presence of myelinating Schwann cells. Since their emergence anticipates that of heminodes, we describe them as early clusters. Unlike heminodes, which are associated with Schwann cell microvilli, nucleation of early clusters is not spatially restricted to sites of any apparent axo-glial structural specialisation. Furthermore, nodal proteins are recruited to early clusters by lateral diffusion in the plane of the axonal membrane and not by direct fusion of vesicular transport vesicles.

Nucleation of early clusters in PNS neurons in co-culture requires neuronal neurofascin 186 (Nfasc186) but is independent of the establishment of axo-glial junctions mediated by the interaction of Caspr with glial neurofascin 155. This is consistent with the requirement of either the glial or the neuronal isoforms of the *Nfasc* gene for Nav recruitment to PNS nodes of Ranvier (*Amor et al., 2017*; *Sherman et al., 2005*; *Zhang et al., 2015*) but contrasts with the fact that neurofascins are neither required for the formation of early clusters in the CNS, nor is neuronal Nfasc an obligate component of these early clusters (pre-nodes) (*Freeman et al., 2015*). Early clusters in PNS neurons in co-culture are mobile and highly dynamic and can disintegrate or fuse with each other. The fact that their number peaks at early stages of myelination and then gradually declines as the more stable heminodes appear suggested a developmental relationship between the two structures. This view is supported by the ability of these mobile early clusters to fuse with neighbouring heminodes.

We show that dynamic early clusters of nodal proteins in PNS nerves represent a Nfasc186-dependent developmental stage of node assembly during myelination that precedes heminode formation. These data underline the uniquely essential role of the *Nfasc* gene in node assembly in the PNS and the diverse strategies that have evolved to ensure the assembly of an essential domain in the vertebrate peripheral nervous system.

## Results

### SEP-Nfasc186 and β1Nav-EGFP are targeted to the axonal plasma membrane during myelination by Schwann cells and assemble into nodal protein clusters

We examined the properties and fate of early clusters in DRG neurons derived from transgenic mice co-cultured with WT rat Schwann cells. The transgenes were expressed under the control of a neuron-specific promoter (*Caroni, 1997*) and encoded either a fusion protein with super-ecliptic pHluorin (SEP) in the extracellular domain of the major neuronal isoform of neurofascin (SEP-Nfasc186) (*Ghosh et al., 2020*) or a fusion of EGFP to the C-terminus of the β1 subunit of Nav, the product of the *SCN1B* gene (β1Nav-EGFP) (*Booker et al., 2020*). SEP is a pH-sensitive GFP derivative that allows selective visualisation of surface-expressed Nfasc186 (*Ashby et al., 2004*; *Ashby et al., 2006*; *Hildick et al., 2012*; *Makino and Malinow, 2009*; *Martin et al., 2008*; *Wilkinson et al., 2014*; *Ghosh et al., 2020*). The transgenic lines are referred to as β1Nav-EGFP and SEP-Nfasc186, respectively, and neurons expressing these proteins are described as being β1Nav-EGFP[+] or SEP-Nfasc186[+]. In order to optimally image myelination in co-cultures of DRG neurons by WT rat Schwann cells, we utilised a microfluidic culture system.

First, we showed that SEP-Nfasc186 and β1Nav-EGFP are targeted appropriately to the axonal membrane during myelination (*Figure 1A and B*). Before myelin ensheathment, both proteins were diffusely distributed along the axolemma but became focally distributed at nodes after myelin ensheathment (*Figure 1A and B*). Early clusters were characteristically flanked by diffuse fluorescent signal in the axonal membrane devoid of adjacent myelin from either SEP-Nfasc186 or β1Nav-EGFP (*Figure 1A and B*). This permitted their ready identification and discrimination from heminodes and nodes by live imaging as myelination proceeded. Early clusters containing fluorescently tagged Nfasc186 or β1Nav are nodal protein complexes (*Figure 1—figure supplement 1A and B*). Hence, the transgenic fusion proteins are appropriate surrogate markers for detecting early clusters, heminodes, and nodes by live imaging.

Similar early clusters have been observed previously in vitro and in vivo, and we have confirmed that they comprise nodal protein complexes in WT co-cultures and WT sciatic nerves (*Figure 1—figure supplement 2A, B and C*; *Eshed-Eisenbach et al., 2020*; *Lambert et al., 1997*). Images of early clusters in WT, β1Nav-EGFP, and SEP-Nfasc186 mice in vivo at P1 (day of birth) are shown

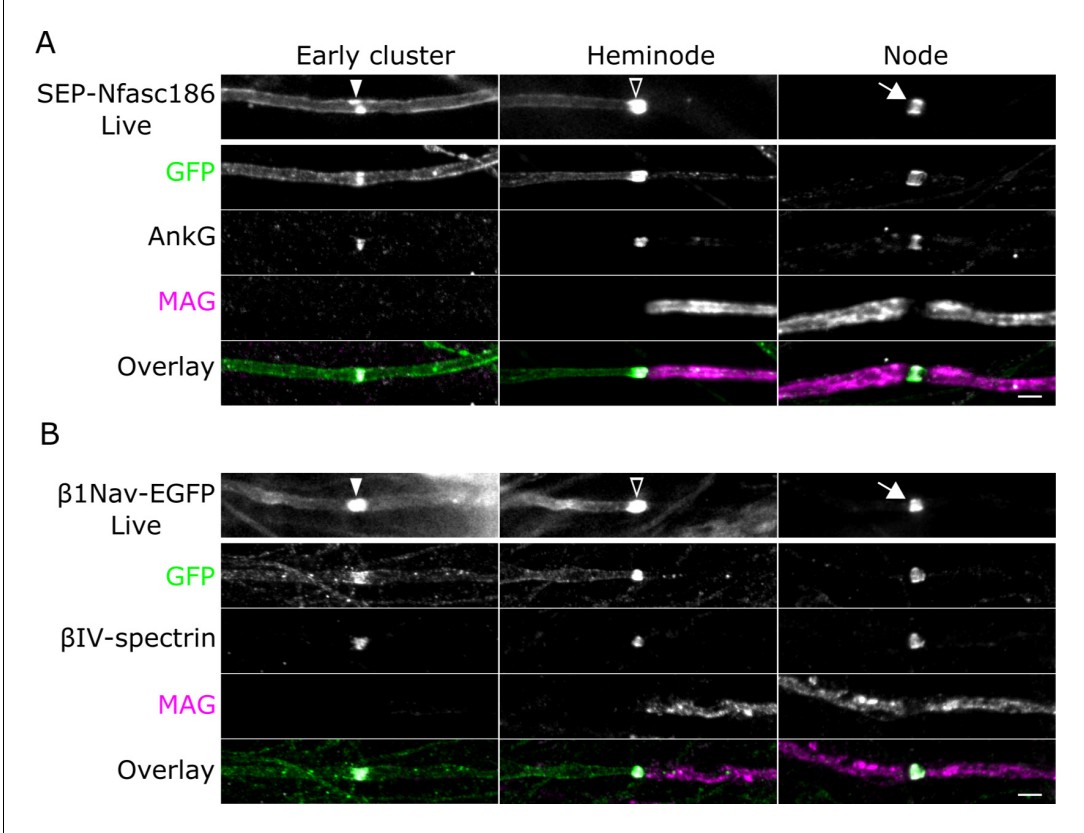

**Figure 1.** SEP-Nfasc186 and β1Nav-EGFP are targeted to the axonal plasma membrane during myelination and assemble into early clusters. (A, B) Live imaging and immunostaining of SEP-Nfasc186[+] (A) or β1Nav-EGFP[+] (B) axons in dorsal root ganglia (DRG) neuron-Schwann cell co-cultures. Both proteins are targeted to early clusters (solid arrowheads) flanked by diffuse signal at the axolemma on unmyelinated axons. SEP-Nfasc186 and β1Nav-EGFP are targeted to heminodes (open arrowheads) and nodes (white arrows) and cleared from the internodal axonal membrane. Immunostaining for GFP identifies SEP-Nfasc186 or β1Nav-EGFP; myelin-associated glycoprotein (MAG) is a component of ensheathing Schwann cells and AnkyrinG (AnkG) and βIV-spectrin are nodal proteins. Scale bars, 3 μm.

The online version of this article includes the following figure supplement(s) for figure 1:

**Figure supplement 1.** SEP-Nfasc186 (A) and β1Nav-EGFP (B) in early clusters (arrowheads) colocalise with nodal proteins.

**Figure supplement 2.** Early clusters assemble in WT axons in vitro and in WT, β1Nav-EGFP, and SEP-Nfasc186 mice in vivo.

(*Figure 1—figure supplement 2C*), and the number of early clusters (without paranodal Caspr) is quantitated as a percentage of the total number of nodal clusters for each mouse line (WT, 86.7 ± 0.9; β1Nav-EGFP, 85.5 ± 1.1; SEP-Nfasc186, 81.7 ± 3.1; ≥145 clusters per mouse, n = 3 mice per genotype, analysis of variance (ANOVA) comparison, not significant). These percentages of early clusters are comparable to those found previously for mice on the day of birth (*Eshed-Eisenbach et al., 2020*).

## Assembly of early clusters is dependent on neuronal neurofascin

The major neuronal isoform of neurofascin, Nfasc186, plays a critical role in the assembly of the node of Ranvier, both in the CNS and in the PNS (*Sherman et al., 2005*; *Zonta et al., 2008*; *Amor et al., 2017*; *Desmazieres et al., 2014*; *Susuki et al., 2013*; *Hedstrom et al., 2007*). A proteolytically processed form of the Schwann cell protein gliomedin interacts with Nfasc186 at heminodes and is also reportedly essential for the assembly of early clusters in the PNS (*Eshed et al., 2005*; *Eshed et al., 2007*; *Maertens et al., 2007*; *Labasque et al., 2011*; *Eshed-Eisenbach et al., 2020*).

Hence, we asked if the interaction between neurofascin in the axonal membrane and gliomedin might account for the ability of gliomedin to promote early cluster assembly.

We compared the number of β1Nav-EGFP$^+$ early clusters in WT and neurofascin-null ($Nfasc^{-/-}$) DRG neurons co-cultured with WT Schwann cells for 10 days after the induction of myelination. Although β1Nav-EGFP$^+$ early clusters assembled in $Nfasc^{+/+}$ neurons, they were barely detectable in $Nfasc^{-/-}$ neurons (β1Nav-EGFP$^+$ early clusters/100 neurons, mean ± SEM: $Nfasc^{+/+}$, 25.40 ± 5.60; $Nfasc^{-/-}$, 0.21 ± 0.05; p = 0.0107, two-tailed unpaired student's t-test, n = 3 independent cultures). These co-cultures are not completely neurofascin-null since WT Schwann cells express the glial neurofascin isoform, Nfasc155. However, this isoform clearly plays no part in early cluster formation. Hence, neuronal neurofascin is not only important in node formation but also necessary for the assembly of early clusters.

## Early clusters assemble in the absence of axo-glial specialisations

The Schwann cell protein gliomedin is not only present at early clusters (*Figure 1—figure supplement 1A and B*) but also required for early cluster formation (*Eshed-Eisenbach et al., 2020*). Therefore, we wished to explore the dependency of early cluster formation on Schwann cells. We showed that not only were Schwann cells required for early cluster formation but also they must be induced to myelinate in the presence of ascorbic acid (*Figure 2A*). Laminin, a component of matrigel, stimulates myelination and enhanced early cluster formation (*Figure 2A*). Hence, the presence of Schwann cells alone is not sufficient for early cluster formation, but they must also be myelinating.

Under myelinating conditions, periaxin-positive Schwann cells were in close proximity to early clusters (*Figure 2B*). However, although gliomedin is also found in the microvilli at the extremities of myelinating Schwann cells (*Eshed et al., 2005*), no other Schwann cell microvillar markers, including pERM (*Figure 2—figure supplement 1A*), dystrophin isoform 116 (Dp116), and radixin (*Figure 2—figure supplement 1B*), were detectable at early clusters, despite being strongly enriched at heminodes and nodes. Thus, early clusters are not contacted by Schwann cell microvilli.

To investigate the structural nature of the proximity of Schwann cells and axons at the ultrastructural level, we performed correlative light and electron microscopy (CLEM) (*Booth et al., 2019*). First, SEP-Nfasc186$^+$ early clusters in DRG neuron-Schwann cell co-cultures were located and imaged live in microfluidic chambers. After fixation, staining, and embedding in resin, regions containing selected early clusters were re-identified, and serially aligned electron microscopy (EM) sections were acquired and imaged using a serial block-face scanning electron microscope (SBF-SEM) (*Figure 2C* and *Figure 2—figure supplement 2*). CLEM confirmed the close contact between Schwann cells and axons at early clusters (*Figure 2C* and *Figure 2—figure supplement 2*). These results underscored the close physical proximity of Schwann cells and axonal early clusters but did not reveal any axo-glial ultrastructural specialisations that might contribute to their formation.

## Early clusters are dynamic and exchange proteins with the surface pool by lateral diffusion

Mature nodes are very stable structures (*Zonta et al., 2011*; *Zhang et al., 2012*). In contrast, heminodes and nascent nodes can exchange proteins with an axolemmal pool (*Zhang et al., 2012*; *Zhang et al., 2020*). The absence of specialised axo-glial structures at early clusters distinguishes them from heminodes and nodes, and prompted us to ask if these structural differences were reflected in the dynamics of these different nodal protein complexes, which might permit them to contribute to nodal development. We assessed the dynamics of nodal protein clusters by fluorescence recovery after photobleaching (FRAP) in SEP-Nfasc186$^+$ DRG neuron-Schwann cell co-cultures. SEP-Nfasc186 expressed on the axons of neurons cultured without Schwann cells was highly mobile (*Figure 3A–C*; *Video 1*), which was consistent with previous studies (*Zhang et al., 2012*; *Ghosh et al., 2020*). SEP-Nfasc186 in early clusters was less mobile, but the recovery of their fluorescence intensity levels indicated that they are highly dynamic structures, whereas nodal SEP-Nfasc186 was relatively immobile (*Figure 3A–C*). SEP-Nfasc186 in heminodes displayed mobilities intermediate between early clusters and nodes (*Figure 3A–C*). Thus, the mobility of SEP-Nfasc186 at nodal clusters gradually declines as it acquires more mature features.

Our observation that heminodes are dynamic structures is particularly intriguing and represents the first direct demonstration that, unlike nodes, they can rapidly exchange full-length Nfasc186 with

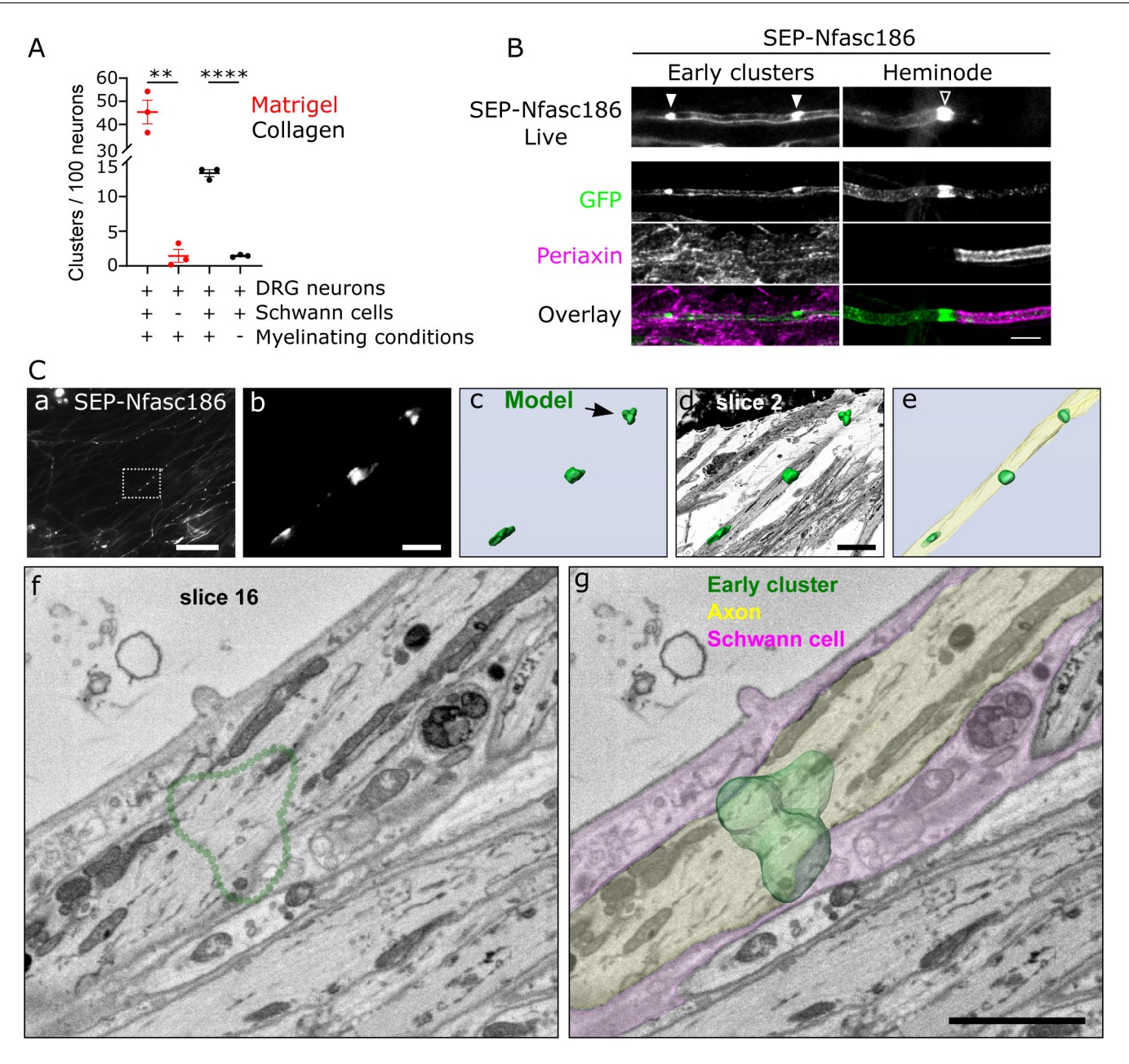

**Figure 2.** Early cluster assembly requires myelinating conditions and occurs close to Schwann cells independently of clear axo-glial membrane specialisations. (A) SEP-Nfasc186[+] dorsal root ganglia (DRG) neurons were grown in matrigel- or collagen-coated microfluidic chambers for 10 days with or without rat Schwann cells, and with myelinating medium or C-medium for further 10 days. Early cluster quantitation: n = 3 independent cultures, unpaired two-tailed Student's t-test, **p<0.01, ****p<0.0001. (B) Schwann cells (immunostained for periaxin) surround an axon with early clusters (solid arrowheads) in myelinating co-cultures. Myelin at heminodes (open arrowhead) was immunostained with periaxin antibodies. Scale bar, 5 µm. (C) Early clusters are in close proximity to Schwann cells. (a) Early clusters in a SEP-Nfasc186[+] co-culture imaged 10 days after myelination induction (scale bar, 50 µm). (b) Magnification of the box in (a) (scale bar, 5 µm). (c) 3D model of the early clusters shown in (b). Arrowhead shows the cluster viewed by electron microscopy (EM) (scale bar, 5 µm). (d) The early cluster model shown in (c) is superimposed on an EM slice of an axon and (e) on a 3D reconstruction obtained from serial EM sections of the same axon. (f) EM section through the early cluster indicated by the arrow in (c). The dotted green line denotes the position of the early cluster. (g) 3D model of the early cluster (dark green) is superimposed on the same EM section shown in (f). A Schwann cell (magenta) and an axon (pale green) are pseudocoloured to show their close proximity. Scale bar, 2 µm.

The online version of this article includes the following source data and figure supplement(s) for figure 2:

**Source data 1.** Source data for *Figure 2*.

**Figure supplement 1.** Early clusters are not contacted by Schwann cell microvilli.

*Figure 2 continued on next page*

*Figure 2 continued*

**Figure supplement 2.** Imaging of early clusters by CLEM.

the axonal pool. We confirmed the dynamic nature of early clusters and heminodes by performing FRAP on the axons of live intercostal nerves in SEP-Nfasc186$^+$ triangularis sterni explants. As observed in the microfluidic co-culture system, SEP-Nfasc186 was highly diffusible in naked, unmyelinated axonal regions, completely immobile at mature nodes, and displayed partial and decreasing mobility at early clusters and heminodes (*Figure 3—figure supplement 1A–C*). Taken together, these experiments demonstrate that both early clusters and heminodes are dynamic structures.

To address how fluorescence recovery of photobleached nodal complexes occurred, we employed FRAP combined with fluorescence loss in photobleaching (FLIP) (FRAP/FLIP) (*Hildick et al., 2012*). It has been reported that Nfasc186 is delivered to nascent nodes at an early stage of node formation by redistribution from an existing surface pool. In contrast, after mature node formation, the protein is delivered by direct vesicular insertion (*Zhang et al., 2012*). To discriminate between these possible routes of fluorescence recovery, we applied the technique of FRAP/FLIP. We have previously validated this method in studying the accumulation of membrane proteins at the axon initial segment (*Ghosh et al., 2020*). Immediately after photobleaching, two regions of interest (ROIs) were positioned on the axonal segments flanking early clusters: these were repeatedly photobleached, thus preventing fluorescence recovery at early clusters by diffusion of SEP-Nfasc186 from the neighbouring unbleached axonal regions. Since vesicular SEP-Nfasc186 is unaffected by repeated photobleaching (*Ghosh et al., 2020*), direct vesicle fusion could, in principle, contribute to fluorescence recovery at early clusters.

FLIP completely prevented FRAP at early clusters (*Figure 3D,E and G*; *Video 2*), indicating that SEP-Nfasc186 fluorescence recovery at these structures is exclusively via lateral diffusion from the axonal surface pool. Interestingly, FRAP/FLIP showed that the same mode of fluorescence recovery applies to heminodes and that this recovery is predominantly mediated by the pool of SEP-Nfasc186 on the flanking unmyelinated axon (*Figure 3D,F and G*).

To control for any FLIP-induced photodamage to the axonal microtubule network that might potentially influence normal vesicular trafficking through the target axonal region, we quantitated axonal vesicular transport in β1Nav-EGFP$^+$ DRG neuron-Schwann cell co-cultures both before and after FLIP (*Figure 3—figure supplement 2A*). We found that neither the number nor the speed of mobile axonal β1Nav-EGFP$^+$ vesicles is affected by FLIP (*Figure 3—figure supplement 2B and C*), indicating that FLIP does not affect microtubule-based vesicle transport of this membrane protein.

To establish if the dynamic nature of SEP-Nfasc186 at early clusters also applied to other components of this nodal protein complex, we used FRAP to analyse the mobility of β1Nav-EGFP at early clusters. In β1Nav-EGFP$^+$ axons, unlike those expressing SEP-Nfasc186, both the intracellular and the cell-surface population of β1Nav-EGFP are visualised by fluorescence microscopy. Therefore, we compared the recovery of the fluorescent signal at early clusters to that of an immediately adjacent axonal segment containing no detectable β1Nav-EGFP$^+$ clusters, which was not subject to FRAP and which we refer to as the control ROI. Recovery of the fluorescent signal in β1Nav-EGFP$^+$ early clusters was not as rapid as for SEP-Nfasc186 (data not shown). However, by 4 hr after photobleaching, β1Nav-EGFP$^+$ early clusters recovered to a level comparable to SEP-Nfasc186 (*Figure 3—figure supplement 3A and B*). Thus, like SEP-Nfasc186, β1Nav-EGFP in early clusters is mobile, but its turnover rate is considerably slower.

In contrast to Nfasc186, delivery of Nav to nascent PNS nodes is thought to occur via direct vesicular insertion (*Zhang et al., 2012*). Whether this is also the case for the Nav accessory protein β1Nav is unknown. To ask whether lateral diffusion or vesicular delivery might account for the recovery of β1Nav-EGFP fluorescence signal at early clusters, we performed FRAP on β1Nav-EGFP$^+$ early clusters in the presence of the microtubule-disrupting agent nocodazole. Nocodazole treatment resulted in a marked reduction in the mobility of β1Nav-EGFP$^+$ vesicles in DRG axons (*Figure 3—figure supplement 3C and D*) but had no effect on the recovery of β1Nav-EGFP$^+$ early clusters after photobleaching (*Figure 3—figure supplement 3E and F*). This indicated that, like Nfasc186, β1Nav-EGFP fluorescence in early clusters recovers by lateral diffusion in the axonal membrane. To exclude the possibility of a generalised reduction in the axonal expression level of β1Nav-EGFP after

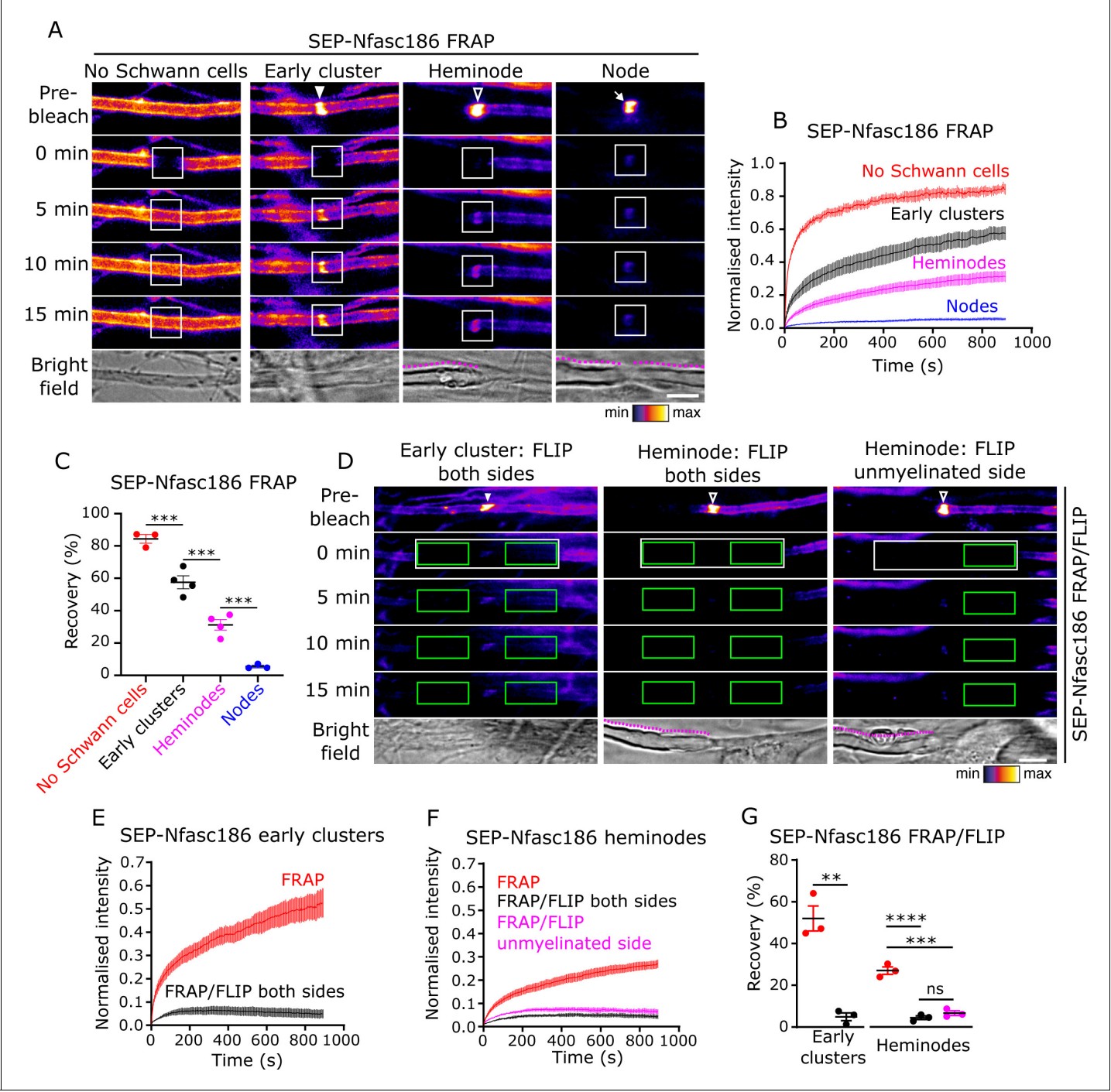

**Figure 3.** Early clusters are dynamic and exchange SEP-Nfasc186 with the surface pool by lateral diffusion in DRG neuron-Schwann cell co-cultures. (**A**) Still video images of fluorescence recovery after photobleaching (FRAP) of a naked axon, an early cluster (solid arrowhead), a heminode (open arrowhead), and a node (arrow). FRAP regions of interest (ROIs) are in white boxes. Magenta lines outline myelin in bright-field images. (**B, C**) The fluorescent signal at unmyelinated axons, early clusters, and heminodes in vitro recovers after photobleaching. n≥3 independent cultures, ≥10 axons per culture, one-way analysis of variance (ANOVA) followed by Sidak's multiple comparisons test; ***p<0.001. (**D**) Still video images of FRAP with bilateral fluorescence loss in photobleaching (FLIP) at an SEP-Nfasc186[+] early cluster (solid arrowhead) and heminode (open arrowhead) and asymmetric FLIP at an SEP-Nfasc186[+] heminode (FRAP ROIs in white boxes and FLIP ROIs in green boxes). (**E–G**) Signal recovery at early clusters and at heminodes is by lateral diffusion. n≥3 independent cultures, ≥10 axons per culture; early clusters: unpaired two-tailed Student's t-test. Heminodes: one-way ANOVA followed by Sidak's multiple comparisons test; **p<0.01; ***p<0.001; ****p<0.0001; ns = not significant. Scale bars, 5 μm. The online version of this article includes the following source data and figure supplement(s) for figure 3:

*Figure 3 continued on next page*

*Figure 3 continued*

**Source data 1.** Source data for *Figure 3*.
**Figure supplement 1.** SEP-Nfasc186 in early clusters is dynamic in the intercostal nerves of triangularis sterni explants.
**Figure supplement 1—source data 1.** Source data for *Figure 3—figure supplement 1*.
**Figure supplement 2.** Vesicular transport in DRG axons is unaffected by FRAP/FLIP.
**Figure supplement 2—source data 1.** Source data for *Figure 3—figure supplement 2*.
**Figure supplement 3.** Delivery of β1Nav-EGFP to early clusters is dynamic and independent of microtubule-based vesicular transport.
**Figure supplement 3—source data 1.** Source data for *Figure 3—figure supplement 3*.

a prolonged nocodazole treatment, we measured the fluorescence intensity in the unbleached control ROI immediately before, immediately after, and 4 hr after photobleaching, and found no significant reduction in fluorescence intensity (*Figure 3—figure supplement 3G*).

Hence, early clusters are dynamic structures capable of exchanging nodal proteins with the axonal surface pool via lateral diffusion.

## Axo-glial junctions restrict the dynamics of nodal complexes

Early clusters contain the known major proteins of mature nodes, but their highly dynamic nature is in stark contrast to the marked stability of nodes and, to a lesser degree, to that of heminodes. Since distinct axo-glial specialisations are not apparent at the ultrastructural level in their vicinity (*Figure 2C*), we predicted that fluorescence recovery of SEP-Nfasc186 in early clusters would be unaffected in Caspr-null DRG axons from *Cntnap1$^{-/-}$* mice (*Figure 4A and B*). Furthermore, the absence of Caspr, and hence of paranodal axo-glial junctions (*Bhat et al., 2001*), did not influence the assembly of early clusters (SEP-Nfasc186$^{+}$ early clusters/100 neurons; mean ± SEM: *Cntnap1$^{+/+}$*, 90.2 ± 28.3; *Cntnap1$^{-/-}$*, 90.9 ± 15.0; p = 0.9838; two-tailed paired student's t-test; n = 3 independent cultures). Perhaps surprisingly, loss of paranodal axo-glial junctions had no significant effect on the mobility of SEP-Nfasc186 in heminodes, but, and consistent with earlier reports (*Zhang et al., 2020*), junctional disruption significantly increased protein mobility at mature nodes (*Figure 4A and B*). The insensitivity of SEP-Nfasc186 mobility in heminodes to junctional disruption presumably reflects the dominant contribution of fluorescence recovery of lateral diffusion from the unmyelinated side that lacks an axo-glial junction (*Figure 3D,F and G*).

These findings were further confirmed by performing FRAP on SEP-Nfasc186$^{+}$ intercostal nerves from triangularis sterni preparations from Caspr-null (*Cntnap1$^{-/-}$*) or Nfasc-null (*Nfasc$^{-/-}$*) animals (*Figure 4—figure supplement 1A–C*).

Paranodal axo-glial junctions have been proposed to stabilise nodes by acting as diffusion barriers (*Zhang et al., 2012*; *Zhang et al., 2020*). Using FRAP and FRAP/FLIP, we now provide direct evidence for this. We observed that the ability of SEP-Nfasc186$^{+}$/Caspr-null nodes to partially recover after photobleaching was completely ablated when fluorescence recovery by lateral diffusion of SEP-Nfasc186 was impeded by performing FRAP/FLIP (*Figure 4—figure supplement 2A–C*). Furthermore, while FLIP alone did not affect the fluorescence intensity of SEP-Nfasc186$^{+}$/WT nodes, it resulted in significant loss of fluorescence at SEP-Nfasc186$^{+}$/Caspr-null nodes (*Figure 4—figure supplement 2D–F*). Collectively, these results support the view that early clusters are highly dynamic structures untrammelled by axo-glial interactions and further demonstrate that the formation of axo-glial junctions has a profound effect on the dynamic properties of mature nodal assemblies.

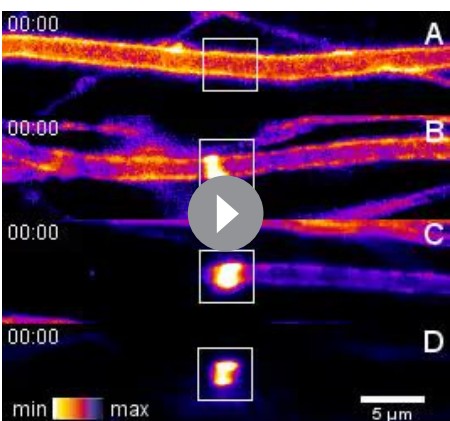

**Video 1.** FRAP (white box) of SEP-Nfasc186 in an axon with no Schwann cells (**A**), early cluster (**B**), heminode (**C**), and node (**D**). Real interframe interval: 4 s.
https://elifesciences.org/articles/68089#video1

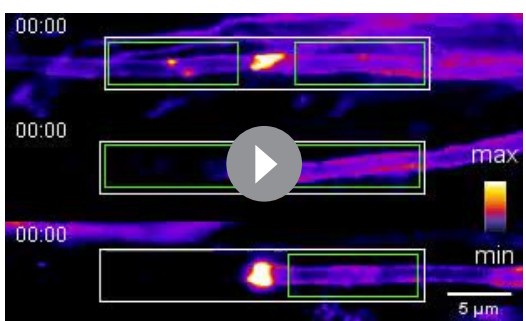

**Video 2.** Bilateral FRAP/FLIP of SEP-Nfasc186 at an early cluster (top panel) and a heminode (middle panels) and unilateral FRAP/FLIP at a heminode (bottom panel). Fluorescence recovery after photobleaching (FRAP) and fluorescence loss in photobleaching (FLIP) regions of interest (ROIs) are delimited by white and green boxes, respectively. Real interframe interval: 4 s.

https://elifesciences.org/articles/68089#video2

## Mobility and migration of early clusters is actin-dependent

Early clusters are highly dynamic structures capable of exchanging proteins with the axonal surface pool. Therefore, we asked whether they are also mobile by performing time-lapse imaging of SEP-Nfasc186[+] early clusters in DRG neuron-Schwann cell co-cultures. Time-lapse imaging of co-cultures 10 days after the induction of myelination showed that early clusters are highly mobile and plastic (*Figure 5*). The vast majority of early clusters remained clearly visible over an 18-hr time-lapse imaging period and appeared to migrate along the axon (*Figure 5A*; *Video 3*). When the position of early clusters was recorded in relation to immobile landmarks, such as axonal branching points or points of intersection with other axons, it became apparent that the perceived movement of early clusters could not be explained by axonal extension (*Figure 5B*). Occasionally, a single early cluster split into two or more smaller clusters that eventually disaggregated (*Figure 5C*; *Video 4*). Interestingly, when multiple early clusters were present on the same axonal segment, they almost invariably moved synchronously in the same direction and at the same speed (*Figure 5D*; *Video 5*).

The fact that we were able to observe heminodes as they fused, and that nodes appeared stable for the duration of the experiment, supported the view that protracted live imaging did not induce cell toxicity (*Figure 5—figure supplement 1*).

To probe the forces that drive this movement, we asked whether the actin cytoskeleton supported the motility of early clusters. To address this, we imaged SEP-Nfasc186[+] early clusters in the presence of the actin-depolymerising agent latrunculin A. Treatment with latrunculin A strongly impaired the movement of early clusters, significantly reducing both the percentage and the speed of early clusters that were mobile (*Figure 5—figure supplement 2A,B and D*). Latrunculin A effectively disrupted the actin cytoskeleton, as evidenced by the dramatic reduction in the intensity of phalloidin staining (*Figure 5—figure supplement 2B*). Having observed that movement of early clusters along axons relies on an intact actin cytoskeleton, we reasoned that myosins, the motor proteins responsible for actin-based motility, might also have a role in early cluster movement. To test this, we recorded early cluster movement in the presence of blebbistatin, a myosin inhibitor specific for myosin II. Blebbistatin significantly reduced the speed of mobile early clusters, indicating that myosin II contributes to early cluster migration along axons (mean ± SEM: dimethyl sulfoxide (DMSO), 0.99 ± 0.09 µm/s; blebbistatin, 0.70 ± 0.03, p = 0.0233, two-tailed paired student's t-test, n = 4 independent cultures, ≥10 axons per culture). These drug treatments affect both axons and Schwann cells in the co-culture system. Therefore, it cannot be excluded that the actin cytoskeleton of both cell types participates in early cluster migration since it is well known that Schwann cells migrate along axons (*Heermann and Schwab, 2013*; *Jessen and Mirsky, 2005*; *Woodhoo and Sommer, 2008*). Indeed, in our co-cultures, Schwann cells are similarly highly motile (data not shown).

## Assembly of early clusters precedes heminode formation

Early clusters are dynamic and mobile and are ultrastructurally distinct from heminodes and nodes. Are they simply ectopic aggregates of nodal proteins or do they have a role in node formation? To address this, we first asked if there is a temporal relationship between these three nodal complexes that might suggest a developmental relationship.

We quantitated the appearance by live imaging of each of these structures between 5 and 40 days after the induction of myelination in the somal compartment of microfluidic devices containing cultures of SEP-Nfasc186[+] DRG neurons with Schwann cells. We consistently observed node-like clusters 5 days after the induction of myelination, a stage at which no heminodes or nodes were

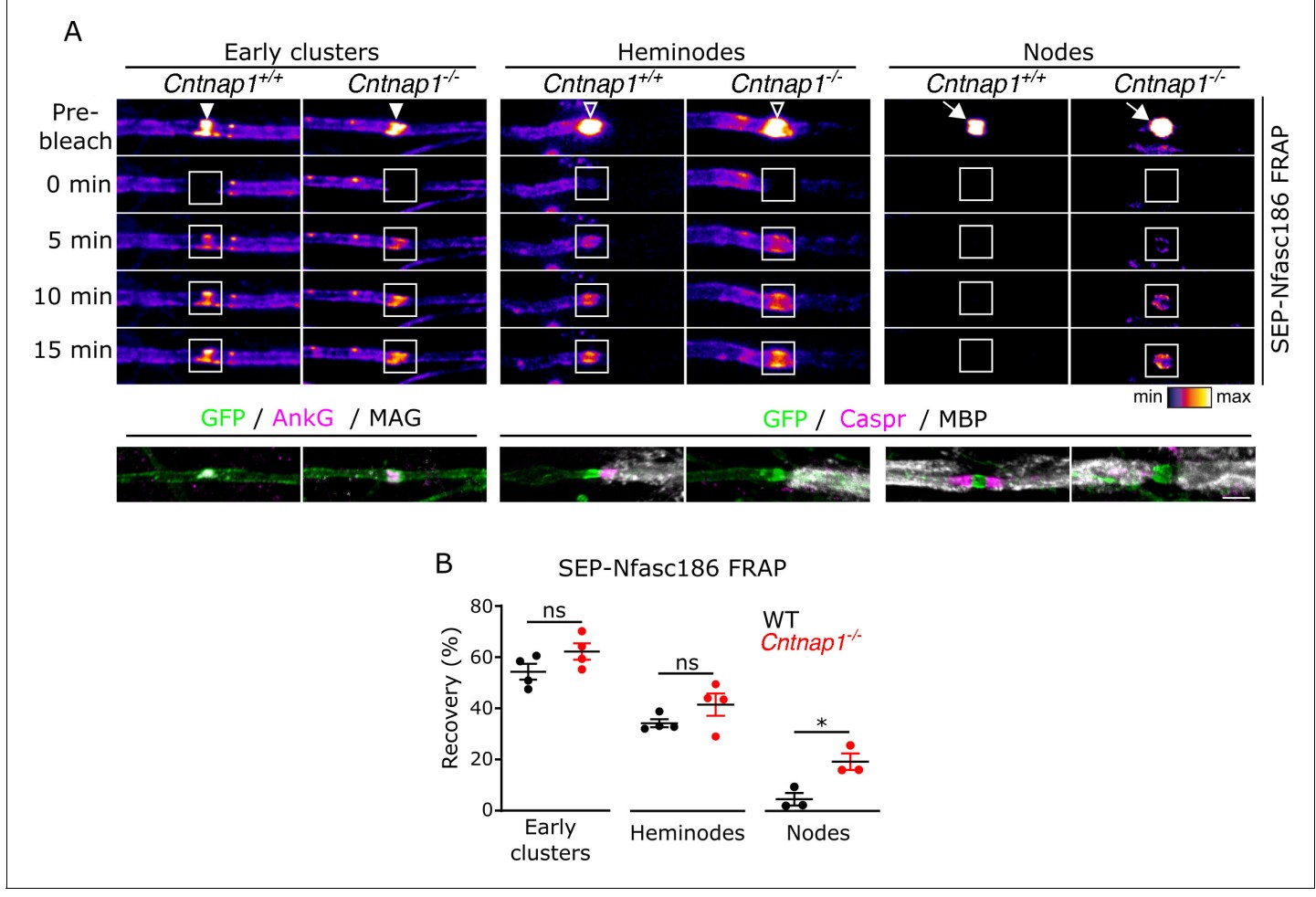

**Figure 4.** Axo-glial junctions restrict the dynamics of nodal complexes. (A) Still video images of fluorescence recovery after photobleaching (FRAP) at SEP-Nfasc186+ early clusters (solid arrowheads), heminodes (open arrowheads), and nodes (white arrows) in *Cntnap1*+/+ or *Cntnap1*-/- dorsal root ganglia (DRG) neurons. FRAP regions of interest (ROIs) are shown as white boxes. (B) The same axons are shown after immunostaining for GFP, AnkyrinG (AnkG), and myelin-associated glycoprotein (MAG) or GFP, Caspr, and MBP (MAG and MBP are myelin markers). Scale bar, 3 μm. (C) Analysis of FRAP curves shows that the fluorescent signal at early clusters and heminodes recovered to the same extent in *Cntnap1*+/+ and *Cntnap1*-/- (Caspr-null) axons 15 min after photobleaching. In contrast, signal recovery after photobleaching was significantly higher at nodes from *Cntnap1*-/- mice. n≥3 independent cultures, ≥10 axons per culture, unpaired two-tailed Student's t-test, *p<0.05.

The online version of this article includes the following source data and figure supplement(s) for figure 4:

**Source data 1.** Source data for *Figure 4*.

**Figure supplement 1.** Axo-glial junctions restrict the dynamics of nodal complexes ex vivo.

**Figure supplement 1—source data 1.** Source data for *Figure 4—figure supplement 1*.

**Figure supplement 2.** Paranodal axo-glial junctions prevent the lateral diffusion of nodal SEP-Nfasc186.

**Figure supplement 2—source data 1.** Source data for *Figure 4—figure supplement 2*.

detectable (*Figure 6A*). The number of early clusters consistently peaked at a time corresponding to the first appearance of heminodes and nodes, and then steadily declined to almost zero by 40 days after the induction of myelination (*Figure 6A*).

Although the appearance of heminodes and nodes in the distal axonal compartment was consistently delayed compared to the somal compartment (*Figure 6A*), we also found, consistent with what we had observed in the somal compartment, that the initiation of assembly of early clusters preceded heminode and node formation in the axonal compartment (*Figure 6A*). Importantly, although we could rarely detect early clusters in the axonal compartment 5 days after the start of myelination (*Figure 6A*), this compartment was already extensively populated by axons at this stage, and Schwann cells were clearly detectable in close proximity to axons (data not shown). This

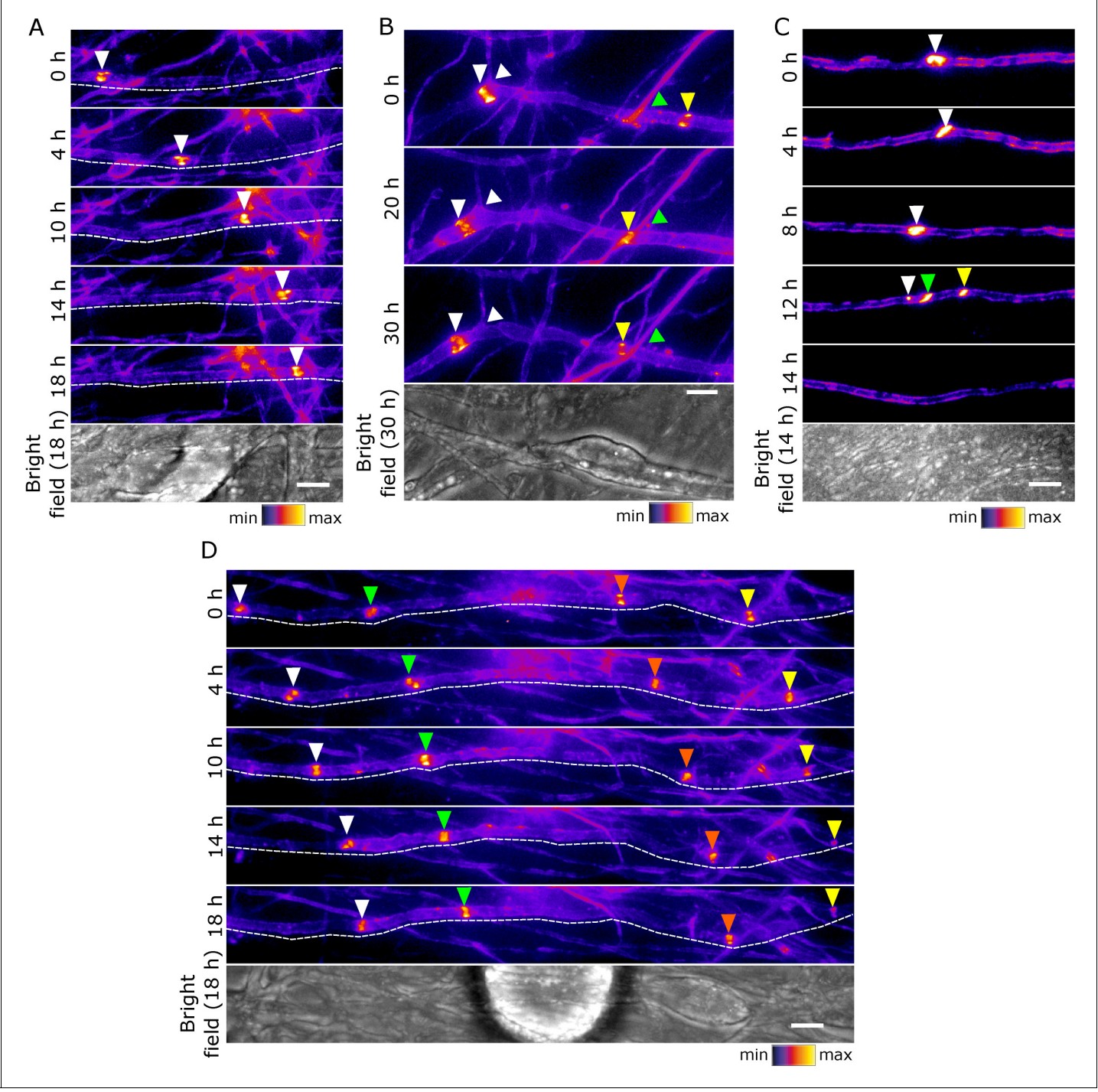

**Figure 5.** Early clusters are mobile and plastic in DRG neuron-Schwann cell co-cultures. (A–D) Still video images of SEP-Nfasc186[+] early clusters at 10 days after the initiation of myelination. Dashed lines outline the imaged axons. Arrowheads of different colours are used to indicate multiple early clusters on the same axon. (A) An individual early cluster migrating along an axon. (B) Two early clusters can be seen moving in the same direction along an axon. One early cluster crosses an axonal branching point (white arrow), while the other moves past a crossing point between axons (green arrow). (C) An early cluster fragmenting and disappearing. (D) A group of early clusters moving synchronously along an axon. Scale bars, 5 μm.

The online version of this article includes the following source data and figure supplement(s) for figure 5:

**Figure supplement 1.** Protracted imaging does not affect myelination.

**Figure supplement 2.** Movement of early clusters is impaired by disruption of the actin cytoskeleton.

**Figure supplement 2—source data 1.** Source data for *Figure 5—figure supplement 2*.

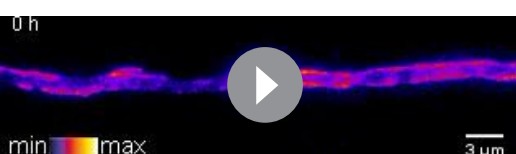

**Video 3.** An SEP-NFASC186[+] early cluster moving along an axon. The video was acquired 10 days after the induction of myelination. Real interframe interval: 2 hr.

https://elifesciences.org/articles/68089#video3

indicates that, like heminodes and nodes, assembly of early clusters proceeds sequentially in a somal-to-distal direction in vitro.

A similar temporal relationship between the sequential appearance of 'isolated' (early) nodal clusters and nodal clusters associated with paranodal Caspr has been noted in vivo (*Eshed-Eisenbach et al., 2020*). However, we refined the temporal analysis of nodal cluster formation in vivo in WT mice to discriminate between heminodes and nodes. We quantitated the percentage of early clusters, heminodes, and nodes at P1, P3, and P5. The sequential appearance of early clusters, followed by heminodes, followed by nodes, observed in co-cultures, was also observed in vivo (*Figure 6B and C*).

Taken together, our data show that early cluster assembly is a temporally regulated developmental stage of PNS myelination both in vitro and in vivo.

## Early clusters contribute to heminodal assembly by fusion

Since early clusters are mobile, we asked if their direction of movement was consistent with their contributing to heminode formation. We quantified early cluster movement 13 days after the start of myelination, when many axons are at least partially ensheathed. To discern whether early clusters preferentially move towards or away from myelinated segments, their movement was recorded distal to the myelinated segment located farthest from the neuronal soma, thus corresponding to the position of the most distal heminode. We found that the vast majority of early clusters were mobile (% of mobile early clusters, mean ± SEM, 84.2 ± 2.2, n = 3 independent cultures, ≥10 axons per culture) and progressed along the axon at an average speed of 0.88 ± 0.03 μm/h (mean ± SEM, n = 3 independent cultures, ≥10 axons per culture). Without exception, mobile early clusters advanced retrogradely towards the nearest heminode, while no early clusters moved in the anterograde direction. Neither the directionality nor the speed of early clusters was influenced by their distance from the nearest myelinated segment (data not shown). Although these studies focussed on distal axonal regions, early clusters in more proximal unmyelinated axonal segments also moved retrogradely in a similar fashion (data not shown).

Early clusters are highly dynamic and mobile. Furthermore, they can fuse with each other (*Figure 7A*; *Video 6*). Therefore, we asked if this plasticity might also allow them to fuse with heminodes. Indeed, we observed that early clusters can not only migrate towards heminodes, but also merge with these structures and so contribute to their assembly (*Figure 7B*; *Video 7*). This is consistent with our earlier observation that heminodes can accrue nodal proteins (and particularly Nfasc186) from the neighbouring axonal surface pool. Collectively, these findings indicate that early clusters can migrate towards and contribute to heminode formation (*Figure 7C*). Hence, they are the earliest nodal complexes in the pathway to the assembly of the node of Ranvier in culture and in vivo (*Eshed-Eisenbach et al., 2020*).

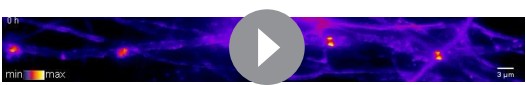

**Video 4.** An SEP-NFASC186[+] early cluster breaking up into three smaller early clusters and eventually disappearing. The video was acquired 10 days after the induction of myelination. Real interframe interval: 2 hr.

https://elifesciences.org/articles/68089#video4

**Video 5.** A group of SEP-NFASC186[+] early clusters synchronously moving along an axon. The video was acquired 10 days after the induction of myelination. Real interframe interval: 2 hr.

https://elifesciences.org/articles/68089#video5

# Discussion

The assembly of specialised domains containing Nav at nodes of Ranvier in the gaps between myelinated segments is essential for the rapid and efficient propagation of action potentials in the CNS and PNS. Disruption of nodal complex assembly is incompatible with normal nervous system development and results in perinatal mortality in rodent models and severe neurodevelopmental defects in humans (*Sherman et al., 2005*; *Maluenda et al., 2016*; *Wambach et al., 2017*; *Laquérriere et al., 2014*; *Lakhani et al., 2017*; *Mehta et al., 2017*; *Conant et al., 2018*; *Nizon et al., 2017*; *Monfrini et al., 2019*; *Smigiel et al., 2018*; *SYNAPS Study Group et al., 2019*). Hence, given their vital role, it is perhaps unsurprising that studies on the node of Ranvier have revealed significant redundancy in the mechanisms that determine its assembly. In this work, we expand the repertoire of mechanistic redundancy that has characterised the evolution of saltatory conduction in vertebrate PNS.

The concept of genetic redundancy, where the protein products of more than one gene can independently fulfil closely related functions, is well-established in biology. In the context of node assembly, this redundancy is illustrated well by the proteins encoded by the *Nfasc* gene. Studies on the glial and neuronal Nfasc isoforms in the PNS have revealed alternate pathways to node assembly. When expressed in Nfasc-null mice in which nodal complexes cannot assemble, the neuronal isoform Nfasc186 has the ability to cluster Nav at PNS nodes in the absence of the glial isoform, Nfasc155 (*Sherman et al., 2005*; *Zonta et al., 2008*). Furthermore, the converse is true: when glial Nfasc155 and its paranodal adhesion complex with contactin and Caspr are intact, nodes assemble in the absence of neuronal Nfasc186 (*Amor et al., 2017*). This represents an unusual, and possibly unique, example of functional redundancy mediated by distinct cell-type-specific proteins encoded by the same gene that have complementary but independent functions in the assembly of the same essential biological structure.

Redundant mechanisms in PNS node assembly can also be identified in the pathways to the formation of a mature node. Heminodal clustering of nodal complexes is widely considered to be the initial step in PNS nodal assembly (*Rasband and Peles, 2021*). Heminodal complexes form at the tips of the microvillar structures located at the extremities of myelinating Schwann cells (*Ching et al., 1999*; *Schafer et al., 2006*). Accumulation of the Schwann cell microvillar protein gliomedin drives the focal recruitment of neuronal Nfasc186, and in turn, AnkG, βIV-spectrin, and Nav at heminodes (*Eshed et al., 2005*). Subsequent convergence and fusion of these heminodal complexes gives rise to mature nodes.

A second mechanism, described as paranodal restriction, depends on the establishment of axoglial junctions at the edges of forming myelin sheaths via the interaction of glial Nfasc155 and neuronal Caspr/contactin (*Rasband and Peles, 2021*). Axo-glial junctions are proposed to act as advancing diffusion barriers that restrict the localisation of axolemma-expressed Nav to the gaps between growing myelin segments (*Rasband and Peles, 2016*). As adjacent paranodal axo-glial junctions converge, they are believed to gradually concentrate Nav in the diminishing gap between them, ultimately resulting in their focal accumulation at nodes (*Rasband and Peles, 2016*).

These two mechanisms appear to act independently to support node formation. Genetic ablation of gliomedin suppresses heminode formation (*Feinberg et al., 2010*), while deletion of either Caspr, Nfasc155, or contactin prevents the establishment of paranodal axo-glial junctions (*Pillai et al., 2009*; *Bhat et al., 2001*; *Boyle et al., 2001*). Nevertheless, neither insult is sufficient to prevent peripheral node formation: when one mechanism fails, the other can rescue node assembly. Mechanisms of node assembly are similarly redundant in the CNS. There, both paranodal axo-glial junctions and specialised interactions between nodal extracellular matrix proteins and Nfasc186 support node formation, and while the former is considered the dominant mechanism, the latter can partially rescue nodal Nav clustering in the absence of axo-glial junctions (*Susuki et al., 2013*; *Amor et al., 2017*).

In this study, we present evidence for the existence of a third mechanism that can contribute to PNS node formation: early clusters of highly mobile nodal protein complexes, which assemble before and independently of heminodes and paranodal axo-glial junctions, actively contribute to the assembly of heminodes by migrating towards and fusing with them. The temporal relationship between early clusters and nodal clusters flanked by paranodal Caspr that we observed in co-culture has also been observed in vivo (*Eshed-Eisenbach et al., 2020*): that is, early clusters appear first,

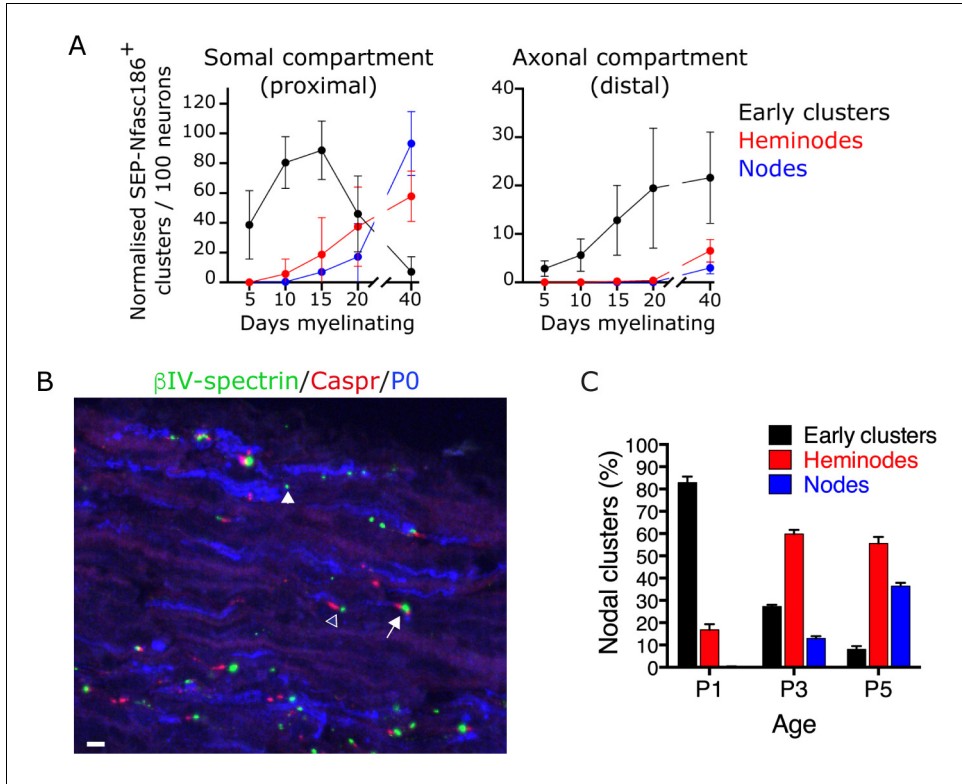

**Figure 6.** Assembly of early clusters precedes heminode formation in co-cultures and in vivo. (**A**) SEP-Nfasc186[+] early clusters, heminodes, and nodes in dorsal root ganglia (DRG) neuron-Schwann cell co-cultures were quantitated at the indicated time points in the somal (proximal) and axonal (distal) compartments of the microfluidic device. n = 3 independent cultures. (**B, C**) Immunostained sections of sciatic nerves from wild-type (WT) mice at P3 for βIV-spectrin (nodal clusters - green), Caspr (paranodes - red), and P0 (myelin - blue). Early clusters (closed arrowhead), heminodes (open arrowhead), and nodes (arrow) were quantitated at P1, P3, and P5, showing the sequential appearance of early clusters, heminodes, and nodes; n≥2 mice. Scale bar, 5 μm.
The online version of this article includes the following source data for figure 6:

**Source data 1.** Source data for *Figure 6*.

and then decline in number in favour of heminodes and nodes. Like heminodes, early cluster assembly depends on the interaction between Schwann cell-secreted gliomedin and neuronal Nfasc186, and also like heminodes, they are not essential for node formation since nodes are still formed in the absence of gliomedin (*Feinberg et al., 2010*). This reinforces redundancy as a consistent feature of node assembly mechanisms.

A recent study has demonstrated that the clustering activity of gliomedin is negatively regulated by proteolytic cleavage by BMP1/Tolloid-like proteases (*Eshed-Eisenbach et al., 2020*). Inhibition of BMP and Tolloid-like proteases results in enhanced formation of early clusters but does not affect myelination or node assembly (*Eshed-Eisenbach et al., 2020*). The BMP1/Tolloid-like proteolytic system was thus suggested to be a mechanism by which Nav clustering is restricted to heminodes during PNS myelination, while the assembly of nodal protein clusters on unmyelinated axons was ascribed to an ectopic epiphenomenon resulting from low levels of gliomedin that had evaded proteolytic degradation and had thus diffused away from sites of active myelination (*Eshed-Eisenbach et al., 2020*).

Several lines of evidence presented in our study support an alternative view. Early clusters can be more appropriately defined as developmental intermediates of node formation. First, we showed that while they are gliomedin-dependent, as demonstrated by others (*Eshed-Eisenbach et al., 2020*), early cluster assembly precedes heminode and node formation, arguing against the possibility that early clusters might originate from excessive active gliomedin escaping the BMP1/TLL1 degradation system and 'spilling over' from neighbouring sites of active node assembly. Interestingly,

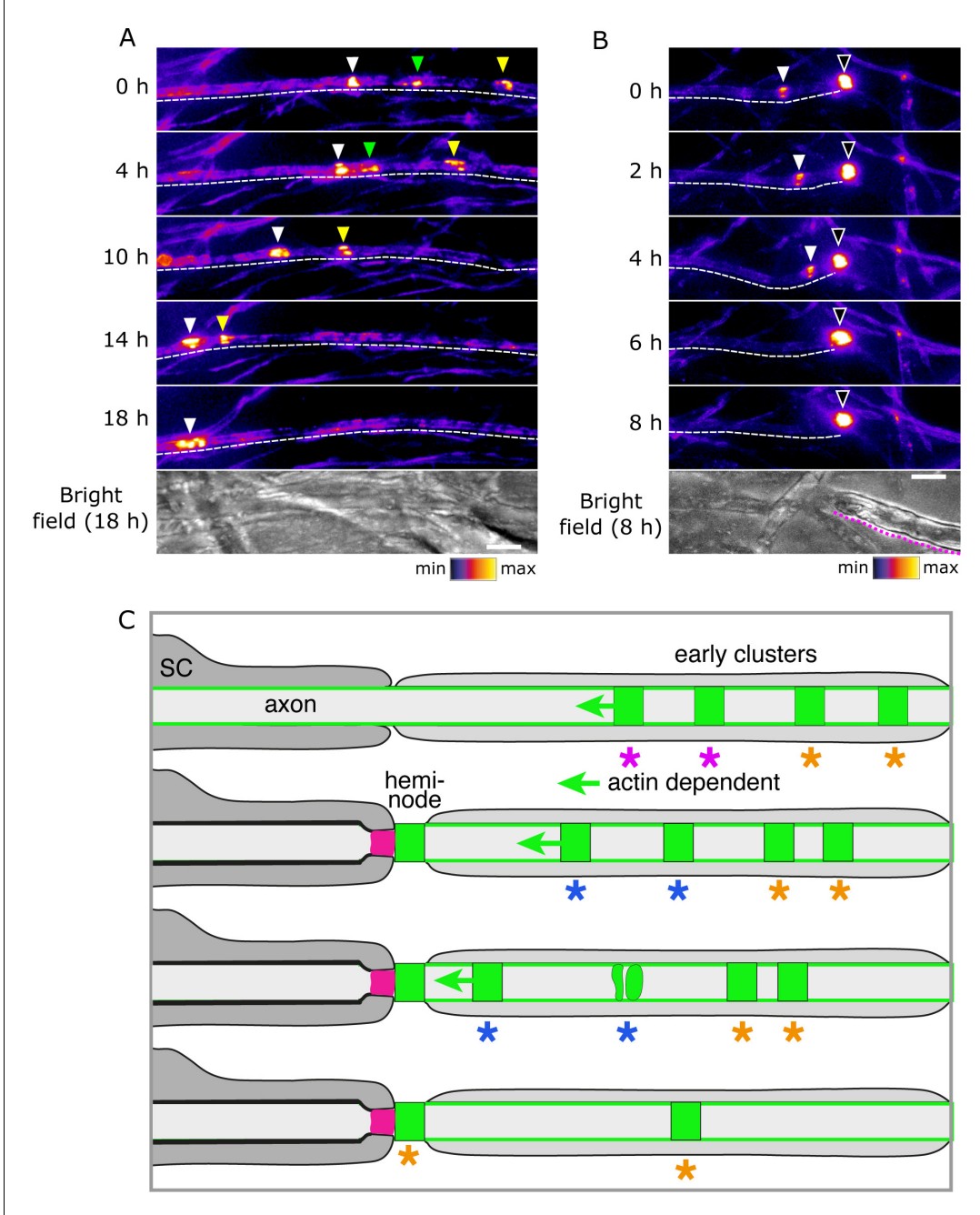

**Figure 7.** Early clusters merge with heminodes. (**A, B**) Still video images of SEP-Nfasc186⁺ early clusters 10 days after myelination was initiated. Dashed lines outline the imaged axons. Dotted magenta line indicates the location of myelin. Arrowheads of different colours are used to indicate multiple early clusters on the same axon. Three early clusters can be seen merging in (**A**). A single early cluster (solid arrowhead) is shown merging with a heminode (open arrowhead) in (**B**). (**C**) Model depicting early clusters migrating, disaggregating, merging, or fusing with a heminode. Key to the colours: dark grey, myelinating Schwann cell with myelin; pale grey, pre-myelinating Schwann cell in contact with the axon; magenta, axo-glial junction; green, nodal complex. Asterisks identify groups of nodal complexes and their fate.

we observed that in our in vitro co-culture system, early cluster assembly was restricted to large cali-bre, A-type Neurofilament (NF200)-positive axons (data not shown), which typically become myelin-ated in vivo (*Lawson et al., 1984*; *Perry et al., 1991*; *Lawson and Waddell, 1991*). Moreover, at later stages of myelination, when several heminodes had already formed but myelination was not yet complete, we could no longer detect early clusters on unmyelinated axons despite their being

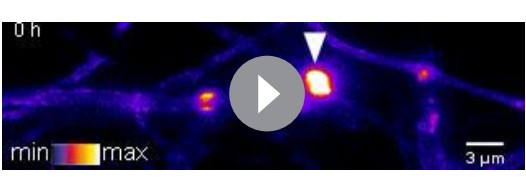

**Video 6.** Three SEP-NFASC186⁺ early clusters merging into one. The video was acquired 10 days after the induction of myelination. Real interframe interval: 2 hr. https://elifesciences.org/articles/68089#video6

extensively populated by Schwann cells (data not shown). This is consistent with the previous observation that in peripheral nerves, early clusters do not assemble on unmyelinated C-type fibres in Remak bundles (*Eshed-Eisenbach et al., 2020*), and supports the view that their formation is restricted to axons that are destined to be myelinated. Second, the peak in the number of early clusters corresponds to the time of first appearance of heminodes and nodes. Subsequently, their number gradually declines as heminodes become more abundant, which supports the existence of a developmental relationship between early clusters and more mature nodal complexes. Third, early clusters migrate towards and fuse with heminodes, thus contributing to their assembly.

The assembly of evenly spaced early clusters of nodal proteins (also referred to as pre-nodes) has been observed in pre-myelinated CNS axons (*Kaplan et al., 1997*; *Kaplan et al., 2001*; *Bonetto et al., 2019*; *Dubessy et al., 2019*; *Freeman et al., 2015*; *Thetiot et al., 2020*). There are many similarities between CNS and PNS early clusters: (i) they assemble before myelination in vitro and in vivo and share a similar molecular composition and size to mature nodes; (ii) they assemble on neurons that typically become myelinated in vivo; (iii) glial-derived secreted factors promote their assembly; (iv) they persist after the onset of myelination; and (v) they enhance axonal conduction velocity (*Kaplan et al., 1997*; *Kaplan et al., 2001*; *Bonetto et al., 2019*; *Dubessy et al., 2019*; *Freeman et al., 2015*; *Thetiot et al., 2020*). It is possible that CNS and PNS early clusters share a similar developmental role. Nevertheless, a clear difference is that assembly of early clusters in the PNS requires neuronal Nfasc186, whereas in the CNS it does not (*Freeman et al., 2015*).

Our study has revealed that PNS early clusters are highly dynamic structures that exchange both Nfasc186 and β1-Nav with the axonal surface pool by lateral diffusion. Nodal protein recruitment to early clusters does not occur by direct vesicular delivery. This extends previous reports indicating that Nfasc186 is recruited to heminodes from an existing axonal surface pool. In contrast, delivery of Nav to nascent nodes reportedly occurs via vesicular transport, but previous studies focussed on the larger, pore-forming α subunit, rather than the smaller, regulatory β subunit of Nav. Genetic ablation of β1-Nav does not prevent the delivery of Nav α subunits to CNS and PNS nodes of Ranvier (*Chen et al., 2004*), demonstrating that the two subunits can be independently delivered to nodes. It is thus possible that the two might follow different delivery routes.

We have provided the first direct demonstration that full-length Nfasc186 is not only delivered to heminodes by lateral diffusion from the flanking unmyelinated axon, but also rapidly exchanges between heminodes at the axonal pool. The ability of both early clusters and heminodes to exchange Nfasc186 with the unmyelinated axon is consistent with their ability to merge.

Our study provides direct evidence that axo-glial structural specialisations, and particularly paranodal axo-glial junctions, determine the extent of lateral mobility of Nfasc186 at nodal complexes. In the absence of axo-glial junctions, mature nodes revert to a more immature state, reacquiring the ability to exchange Nfasc186 with the axonal surface pool by lateral diffusion. This has two important implications: first, it indicates that the strength and nature of the molecular interactions taking place between Nfasc186 and other nodal proteins (e. g., cytoskeletal scaffold proteins) at mature nodes and early clusters are not intrinsically different, but that they are modulated by the presence of the axo-glial complex; second, it reveals that in the absence of axo-glial junctions, Nfasc186 could diffuse through the internodal regions of myelinated axons, and could, conceivably, exchange between adjacent nodes.

An important structural difference between early clusters and more mature nodal complexes that has been revealed by this study is that, despite being closely contacted by periaxin-

**Video 7.** An SEP-NFASC186⁺ early cluster migrating towards and eventually merging with a heminode (solid arrowhead). The video was acquired 10 days after the induction of myelination. Real interframe interval: 2 hr. https://elifesciences.org/articles/68089#video7

positive, myelinating Schwann cells, and despite their focal accumulation of gliomedin, early clusters are not contacted by Schwann cell microvilli, which are known to emerge at early stages of Schwann cell maturation before the start of myelin ensheathment (*Gatto et al., 2003*). The absence of Schwann cell microvilli likely contributes to the highly dynamic properties of early clusters, including their ability to migrate towards and readily merge with heminodes.

What propels the movement of early clusters towards heminodes? Although we have shown that early cluster movement is driven by the acto-myosin complex, we could not discern whether this is Schwann cell- or axon-driven. To our knowledge, and excluding heminodes, this is the first documented example of long-range movement of membrane-bound protein complexes along the axon. Although we cannot exclude an axon-based mechanism, based on our observation that Schwann cells contacting early clusters also appear to migrate along the axon at similar rates (data not shown), we hypothesise that these cells might be responsible for early cluster motility. The mechanisms of early cluster mobility will be the subject of future work.

In conclusion, we describe a new aspect of the diverse redundant mechanisms that appears to be a characteristic of a critical stage in the development of the vertebrate nervous system, namely the assembly of nodes of Ranvier. We have presented evidence that early clusters of nodal protein complexes represent the earliest Nav clustering event in the PNS, that their assembly is at an early developmental stage of myelination, and that they can actively contribute to the assembly of nodes of Ranvier.

## Materials and methods

### Key resources table

| Reagent type (species) or resource | Designation | Source or reference | Identifiers | Additional information |
| --- | --- | --- | --- | --- |
| Strain (*Mus musculus*), C57BL/6JOla, male and female | $Nfasc^{-/-}$ mice | *Sherman et al., 2005* | | Peter Brophy, University of Edinburgh |
| Strain (*Mus musculus*), C57BL/6JOla, male and female | $Cntnap1^{-/-}$ mice | *Gollan et al., 2003* | | E Peles, Weitzmann Institute |
| Strain (*Mus musculus*), C57BL/6JOla, male and female | Thy1-SEP-Nfasc186 | This paper | | Peter Brophy, University of Edinburgh |
| Strain (*Mus musculus*), C57BL/6JOla, male and female | ß1Nav-EGFP | *Booker et al., 2020* | | Peter Brophy, University of Edinburgh |
| Antibody | P0 (chicken polyclonal) | Aves Labs | RRID:AB-2313561 | 1:100 |
| Antibody | Sodium channel (Pan) (mouse monoclonal) | Sigma-Aldrich | Cat# S8809 | 1:200 |
| Antibody | MAG (mouse monoclonal) | M Filbin | IgG1 | 1:50 |
| Antibody | ßIV-spectrin (rabbit polyclonal) | *Desmazieres et al., 2014* | PJ Brophy | 1:200 |
| Antibody | ßIV-spectrin (rabbit polyclonal) | MN Rasband, Baylor College of Medicine | | 1:400 |
| Antibody | AnkyrinG (mouse monoclonal) | UC Davis/NIH NeuroMab | Cat# 75–147 | 1:50 |
| Antibody | GFP (chicken polyclonal) | Abcam | Cat# ab13970 | 1:1000 |
| Antibody | β3-tubulin (mouse monoclonal) | Sigma-Aldrich | Cat# T8660 | 1:100 |
| Antibody | Gliomedin (rabbit polyclonal) | E Peles | | 1:100 |

*Continued on next page*

*Continued*

| Reagent type (species) or resource | Designation | Source or reference | Identifiers | Additional information |
|---|---|---|---|---|
| Antibody | Caspr (rabbit polyclonal) | DR Colman | | 1:5000 |
| Antibody | Caspr (guinea-pig polyclonal) | M Bhat, University of Texas, San Antonio | | 1:400 |
| Antibody | MBP (rabbit polyclonal) | *Vouyioukiis and Brophy, 1993* | PJ Brophy, pep7 | 1:1000 |
| Antibody | Periaxin (rabbit polyclonal) | *Gillespie et al., 1994* | PJ Brophy, Repeat region | 1:5000 |
| Antibody | pERM (rabbit polyclonal) | Cell Signalling Technologies | Cat# 3141 | 1:200 |
| Antibody | Dystrophin (mouse monoclonal) | Sigma-Aldrich (MANDRA1) | Cat# D8043 | 1:200 |
| Antibody | Radixin (rabbit polyclonal) | *Sherman et al., 2012* | P Brophy, RAD4 | 1:500 |
| Antibody | Neurofascin-Pan (rabbit polyclonal) | *Tait et al., 2000* | PJ Brophy, NFC | 1:2000 |
| Antibody | Goat Anti-mouse IgG1 Alexa Fluor 488 | Invitrogen | Cat# A-21121 | 1:1000 |
| Antibody | Goat Anti-rabbit Alexa Fluor 647 | Invitrogen | Cat# A-32733 | 1:1000 |
| Antibody | Donkey Anti-rabbit Alexa Fluor 594 | Jackson ImmunoResearch | Cat# 111-585-14 | 1:1000 |
| Antibody | Donkey Anti-chicken Alexa Fluor 488 | Jackson ImmunoResearch | Cat# 703-545-155 | 1:1000 |
| Antibody | Goat Anti-mouse IgG2a Alexa Fluor 568 | Invitrogen | Cat# A-21134 | 1:1000 |
| Antibody | Goat Anti-mouse IgG2a Alexa Fluor 647 | Invitrogen | Cat# A-21241 | 1:1000 |
| Antibody | Goat Anti-mouse IgG2b Alexa Fluor 488 | Invitrogen | Cat# A-21141 | 1:1000 |
| Antibody | Anti-mouse IgG1 Alexa Fluor 594 | Invitrogen | Cat# A-21125 | 1:1000 |
| Antibody | Goat Anti-mouse IgG1 Alexa Fluor 647 | Invitrogen | Cat# A-21240 | 1:1000 |
| Antibody | Goat Anti-mouse IgG2b DyLight 405 | Jackson ImmunoResearch | Cat# 115-475-207 | 1:100 |
| Antibody | Goat Anti-guinea pig Alexa Fluor 594 | Invitrogen | Cat# A-11076 | 1:8000 |
| Antibody | Goat Anti-rabbit Alexa Fluor 488 | Invitrogen | Cat# 710369 | 1:1000 |
| Antibody | Goat Anti-chicken Alexa-Fluor 647 | Invitrogen | Cat# A-21449 | 1:1000 |
| Chemical compound, drug | Phalloidin AlexaFluor 568 | Invitrogen | Cat# A-12380 | 1:50 |
| Chemical compound, drug | DAPI (4 ′, 6-diamidino-2-phenylindole) | Sigma-Aldrich | Cat# D9542 | 1 µg/ml |
| Chemical compound, drug | BrainStain fluorescent dye mix | ThermoFisher | Cat# B34650 | 1:300 |
| Chemical compound, drug | DMSO | Sigma-Aldrich | Cat# D2650 | |
| Chemical compound, drug | Poly-D-lysine | Sigma-Aldrich | Cat# P6407 | |

*Continued on next page*

*Continued*

| Reagent type (species) or resource | Designation | Source or reference | Identifiers | Additional information |
|---|---|---|---|---|
| Chemical compound, drug | B-27 | Thermo Fisher Scientific | Cat# 17504044 | |
| Chemical compound, drug | Matrigel | Corning | Cat# 356231 | |
| Chemical compound, drug | Fish skin gelatin | Sigma-Aldrich | Cat# G7765 | |
| Chemical compound, drug | Nocodazole | Sigma-Aldrich | Cat# SML1665 | |
| Chemical compound, drug | Latrunculin A | Merck | Cat# 428026 | |
| Chemical compound, drug | para-amino-Blebbistatin | Cayman Chemical | Cat# 22699 | |
| Software, algorithm | FIJI | *Schindelin et al., 2012* | RRID:SCR_002285 | https://imagej.net/Fiji |
| Software, algorithm | Prism 8.0 | GraphPad | RRID:SCR_002798 | |
| Software, algorithm | KymoTool Box | *Zala et al., 2013* | | F Saudou, University Grenoble Alpes |
| Software, algorithm | Amira | ThermoFisher | RRID:SCR_007353 | |

## Animals

All animal work conformed to UK legislation (Scientific Procedures) Act 1986 and to the University of Edinburgh Ethical Review policy. Caspr-null, Nfasc-null, and β1Nav-EGFP mice have been previously described (*Gollan et al., 2003*; *Sherman et al., 2005*; *Booker et al., 2020*). To generate SEP-Nfasc186 transgenic mice, an SEP-Nfasc186 transgene was constructed by inserting a restriction site (Age I) by site-directed mutagenesis in the murine Nfasc186 cDNA (*Zonta et al., 2008*) at amino acid 38 between the signal sequence and the first IgG domain. SEP cDNA (a gift from Dr. Gero Miesenböck, University of Oxford) was cloned into the Age I site and then inserted into a plasmid containing the Thy1.2 promoter (*Caroni, 1997*). Transgenic mice were generated by pronuclear injection as described (*Sherman and Brophy, 2000*). All mice were backcrossed to a C57BL/6 background for at least 10 generations. SEP-Nfasc186 mice were interbred with $Cntnap1^{+/-}$ mice or $Nfasc^{+/-}$ mice to generate SEP-Nfasc186/$Cntnap1^{-/-}$ and SEP-Nfasc186/$Nfasc^{-/-}$ mice. β1Nav-EGFP mice were interbred with $Nfasc^{+/-}$ mice to generate β1Nav-EGFP/$Nfasc^{-/-}$ mice.

## Preparation of microfluidic devices

Microfluidic devices were generated using photo/soft-lithography techniques as previously described (*MacKerron et al., 2017*). Briefly, two-layer microfluidic masters were fabricated by spinning SU8 photoresist (3000 series; Microchem) onto a silicon wafer. The first layer (SU8 3005) created straight microchannels (3-μm thick, 10-μm wide, and 500-μm long), while a second layer (SU8 3035) created the culture chambers (100-μm thick, 2-mm wide, and 9.5-mm long). The silicon masters were silanized by vapour deposition of 1H, 1H, 2H, 2H-perfluorooctyl-trichlorosilane (Sigma-Aldrich) for 1 hr. To fabricate devices, polydimethylsiloxane (PDMS) base silicone elastomer (Sylgard 184; Dow Corning) was mixed with curing agent (10:1 ratio), poured on the silicon master, degassed in a vacuum desiccator chamber, and cured at 80°C overnight. The cured PDMS was then peeled off the silicon master, individual devices were cut to the desired size, and inlet and outlet wells were created using an 8 mm surgical biopsy punch (Stiefel). Unless otherwise specified, the devices were cleaned using vinyl tape and then permanently bonded to glass coverslips (#1.5; Menzel-Glaser) using oxygen plasma surface treatment (Zepto Plasma cleaner; Diener electronics). Plasma-bonded devices were immediately filled with sterile $H_2O$ and placed in sterile 6 cm dishes. After washing three times with $H_2O$, the devices were sterilised by a 5 min UV light treatment in a UV cross-linker (Amplirad), then washed again with sterile $H_2O$, and stored at 4°C until coating. For CLEM, glass coverslips were sterilised by autoclaving, then coated for 1 hr at 37°C with poly-D-lysine (PDL) (30 μg/ml; Sigma-Aldrich) in phosphate-buffered saline (PBS), followed by three washes with $H_2O$, and allowed to air

dry. Autoclaved PDMS devices were then transiently bonded to PDL-coated coverslips by gently pressing the devices onto the glass. Unless otherwise specified, bonded devices were coated for 1 hr at room temperature (RT) with phenol red-free growth factor-reduced matrigel (0.45 mg/ml; Corning) in phenol red-free Dulbecco's modified Eagle medium (DMEM) (Gibco, Thermo-Fisher). Where specified, devices were instead coated for 1 hr at RT with 0.45 mg/ml Rat Collagen I (Cultrex, R and D Systems,) in phenol red-free Hank's Balanced Salt Solution (HBSS) with calcium and magnesium (Sigma-Aldrich). After matrigel or collagen coating, devices were coated for 30 min at RT with PDL (30 µg/ml) in PBS, followed by three washes with $H_2O$. Devices were then filled with DMEM$^+$ (-DMEM with pyruvate supplemented with GlutaMAX (1%), Pen/Strep (1%) (all from Gibco, Thermo-Fisher), and 10% fetal calf serum (FCS) (Thermo-Fisher)) and stored in a 37°C 5% $CO_2$ incubator until neuron seeding.

## DRG neuron-Schwann cell co-cultures

DRG were isolated from neonatal mice (P6-P8) as previously described (*Sleigh et al., 2016*) and collected in ice-cold $Ca^{+2}$- and $Mg^{+2}$-free HBSS (Sigma-Aldrich). DRG were then dissociated for 10 min at 37°C and 5% $CO_2$ with papain (2 mg/ml; Worthington Biochemical) and L-cysteine (0.36 mg/ml; Sigma-Aldrich) in phenol red-, $Ca^{+2}$-, and $Mg^{+2}$-free HBSS, followed by 15 min at 37°C and 5% $CO_2$ in collagenase (8 mg/ml; Invitrogen) in phenol red-, $Ca^{+2}$-, and $Mg^{+2}$-free HBSS. After a brief wash in DMEM$^+$, cells were resuspended in 5 ml DMEM$^+$ and panned for 1 hr at 37°C and 5% $CO_2$ in a 60 mm tissue culture-treated dish to remove most adherent non-neuronal cells, then resuspended in 20 µl of resuspension medium per pup (DMEM$^+$ containing FUDR (20 µM 5-fluoro-2'-deoxyuridine, 20 µM uridine; Sigma-Aldrich), and nerve growth factor (NGF) (50 ng/ml; R and D Systems)). After emptying all inlet and outlet wells of the microfluidic device, 4 µl of the cell suspension was added to the left wells and incubated for 40 min in a 37°C 5% $CO_2$ incubator to allow the neurons to adhere to the substrate. The inlet and outlet wells of the microfluidic device were then filled with resuspension medium before returning the device to the incubator. The day after neuron seeding, the plating medium was replaced with supplemented neurobasal medium (NBS) (phenol red-free neurobasal-A (Gibco, Thermo-Fisher), B27 (1%; Gibco, Thermo-Fisher), D-glucose (4 g/l; Sigma-Aldrich), GlutaMAX (1%), Pen/Strep (1%), NGF (50 ng/ml), and FUDR (20 µM)), and the medium was refreshed every other day until Schwann cell seeding or throughout the life of the culture. At least 2 days before seeding Schwann cells, the neuronal culture medium was replaced with NBS without FUDR. Rat Schwann cells were purified from neonatal sciatic nerves as previously described (*Grove and Brophy, 2014*). Schwann cells were maintained in SC DMEM (DMEM (Gibco, Thermo-Fisher), FCS (10%), GlutaMAX (1%), Pen/Strep (1%)) supplemented with forskolin (2 µg/ml; Sigma-Aldrich) and neuregulin 1 (10 µg/ml; R and D Systems) until 2 days before seeding onto DRG neurons, when they were starved by omission of forskolin and neuregulin 1. When ready for seeding, Schwann cells were harvested in trypsin/ethylenediaminetetraacetic acid (EDTA) (0.25%; Gibco, Thermo-Fisher), washed in SC DMEM, and resuspended in C-medium (phenol red-free MEM (Gibco, Thermo-fisher), FBS (10%), D-glucose (4 g/l), GlutaMAX (1%), Pen/Strep (1%)). Shortly before seeding Schwann cells, the DRG neuron culture medium was replaced with C-medium. After emptying all inlet and outlet wells of the microfluidic device containing the DRG neurons, 4 µl of Schwann cell suspension was pipetted into the left hand culture chamber and incubated for 20 min in a 37°C 5% $CO_2$ incubator to allow Schwann cells to adhere to the substrate. The inlet and outlet wells of the microfluidic device were then filled with C-medium before returning the device to the incubator. At the time of adding Schwann cells, DRG neuron cultures were 9–11 days old. The day after seeding Schwann cells, myelination was initiated by replacing C-medium with myelinating medium (C-medium, NGF (50 ng/ml), ascorbic acid (50 µg/ml, Sigma-Aldrich)).

## Drug treatments

The actin-depolymerising agent latrunculin A (Merck), the microtubule-depolymerising agent nocodazole (Sigma-Aldrich), the myosin II ATPase inhibitor blebbistatin (para-amino-Blebbistatin; Cayman Chemical) or an identical volume of anhydrous DMSO (Sigma-Aldrich) was diluted in myelinating medium. All drug treatments were performed on DRG neuron-Schwann cell co-cultures from 10 days after the start of myelination. Co-cultures were pre-treated for 30 min with 5 µM latrunculin A or DMSO, and then switched to 50 nM latrunculin A immediately before imaging. Nocodazole (30 µM)

was added to co-cultures 30 min before FRAP, and treatment continued for 4 hr after FRAP. Blebbistatin (50 μM) was added to co-cultures immediately before imaging.

## Immunofluorescence and image acquisition

DRG neuron-Schwann cell co-cultures were fixed with 4% methanol-free formaldehyde (Thermo-Fisher) in PBS for 15 min at RT, followed by three washes in PBS. Sciatic nerves (P1) were fixed by immersion in 4% paraformaldehyde (PFA) in 0.1 M sodium phosphate buffer (pH 7.4) for 30 min at RT, followed by three washes in PBS, and teased on Superfrost slides (Thermo-Fisher). Fixed samples were blocked for 30 min in blocking buffer containing 5% fish skin gelatin (Sigma-Aldrich) and 0.2% Triton X-100 (Sigma-Aldrich) in PBS followed by incubation with primary antibodies diluted in blocking buffer at RT overnight. Immunolabelling of cryostat sections of P1 sciatic nerves was performed as described previously (*Tait et al., 2000*). The fluorescently labelled secondary antibodies or Phalloidin Alexa Fluor 568 was diluted in blocking buffer and incubated for 2 hr at RT. Samples were mounted in Vectashield Mounting Medium (Vector Laboratories). When DAPI was used to stain nuclei, it was added to the mounting medium. Representative images of DRG neuron-Schwann cell co-cultures were acquired on a Zeiss LSM710 confocal microscope with a Plan-Apochromat X20 objective (NA 0.8), a Plan-Apochromat X40 oil objective (NA 1.3), and a Plan-Apochromat X63 oil objective (NA 1.4), all from Zeiss, or on a Leica TCS SP8 confocal microscope with a Plan Apochromat X63 oil objective (NA 1.4; Leica). Images of teased sciatic nerve fibres were acquired on a Leica TCL-SL confocal microscope equipped with a X63 objective lens (NA 1.4; Leica) using Leica proprietary software.

## Live-cell imaging of DRG cultures

Live imaging of DRG neurons and DRG neuron-Schwann cell co-cultures was performed using an inverted wide-field microscope (Zeiss AxioObserver), equipped with the following objectives: alpha Plan Apochromat X100 oil DIC (NA 1.46; Zeiss), EC Plan-Neofluar X10 Ph1 (NA 0.30; Zeiss), and Plan-Apochromat X20 M27 Ph2 (NA 0.80; Zeiss), together with Definite Focus.2 (for z-drift correction), a WSB PiezoDrive 08 (Zeiss), an ORCA-Flash4.0 V2 Digital CMOS camera (Hamamatsu Photonics), and a 37 ˚C imaging chamber (PeCon) in a humidified atmosphere containing 5% $CO_2$. LED illumination (Colibri 7; Zeiss) was used for image acquisition and camera pixel size was binned to 2x2 to achieve better signal-to-noise ratios. The entire imaging workflow was controlled by Zeiss imaging software (ZEN 2.3). For photomanipulation, the microscope was coupled to a diode laser (473 nm, 100 W) and a laser scanning device (UGA-42 Firefly; Rapp OptoElectronic). The laser was controlled using Rapp's proprietary SysCon software and synchronized to image acquisition by ZEN 2.3. Unless otherwise specified, DRG neuron-Schwann cell co-cultures were imaged in myelinating medium using a low LED power (15%) with a 100 ms exposure time to minimize phototoxicity. Time series documenting early cluster movement in latrunculin-, blebbistatin-, or DMSO-treated SEP-Nfasc186[+] DRG neuron-Schwann cell co-cultures was acquired at a rate of 1 time point every 2 hr for 16 hr. Each time point consisted in the maximum intensity orthogonal projection of a z-series of 30 focal planes with a z-step size of 240 nm. During live imaging, early clusters in SEP-Nfasc186[+] or β1Nav-EGFP[+] DRG neurons were identified based on the absence of myelin flanking the cluster (evaluated using bright-field illumination and a phase contrast objective) and presence of the fluorescent signal along the axolemma flanking the cluster. Heminodes and nodes were identified based on the presence of myelin on one or both sides of the cluster, respectively, combined with the presence of the fluorescent signal along the unmyelinated axolemma flanking the cluster. After live imaging, the identity of all imaged early clusters, heminodes, and nodes was confirmed by staining for GFP, MBP, and either AnkG or βIV-spectrin.

## Fluorescence recovery after photobleaching

For FRAP experiments, early clusters, heminodes, or nodes in SEP-Nfasc186[+] or β1Nav-EGFP[+] DRG neuron-Schwann cell co-cultures, or comparably sized segments of unmyelinated axons in SEP-Nfasc186[+] DRG cultures, were photobleached using a 473 nm laser (15% power for 15 cycles). Unless otherwise specified, pre-bleach and post-bleach frames were acquired at the rate of 1 frame every 4 s for 1 min 20 s and 15 min, respectively. Where specified, signal in the FRAP ROI in β1Nav-EGFP[+] DRG neuron-Schwann cell co-cultures was allowed to recover for 4 hr before acquiring a

single post-bleach frame. FRAP experiments were performed on early clusters, heminodes, and nodes in DRG neuron-Schwann cell co-cultures after 10, 20 and 30 days from the start of myelination, respectively. FRAP of unmyelinated axons in SEP-Nfasc186[+] DRG cultures was performed when the cultures were 19–21 days old, matching the age of 10 days myelinating co-cultures.

## FRAP-FLIP

The FRAP-FLIP protocol was adapted from the method previously described by *Hildick et al., 2012*. After acquiring pre-bleach frames at the rate of 1 frame every 4 s for 1 min 20 s, a single 25-µm-long rectangular ROI centred on a node in SEP-Nfasc186[+]/WT or SEP-Nfasc186[+]/Caspr-null DRG neuron-Schwann cell co-cultures was photobleached as described above and allowed to recover for 15 min. During this period of recovery, two 10-µm-long rectangular ROIs located on either side of the photobleached node were repeatedly photobleached and imaged at intervals of 4 s. These FLIP ROIs were photobleached using the 473 nm laser (5% power for 1 cycle every 4 s) to achieve effective photobleaching. The distance between the two FLIP ROIs was fixed at 5 µm, thus leaving a 1–2 µm gap between each FLIP ROI and the photobleached node. The FRAP-FLIP experiments were carefully evaluated to ensure cytoskeletal integrity was not compromised by measuring vesicle movement in five β1Nav-EGFP[+] axons immediately before and immediately after FRAP/FLIP. Vesicles were tracked as detailed below.

## FLIP

For FLIP experiments, two 10-µm-long rectangular ROIs located on either side of nodes in SEP-Nfasc186[+]/WT or SEP-Nfasc186[+]/Caspr-null DRG neuron-Schwann cell co-cultures were repeatedly photobleached as described above. Pre-bleach and post-bleach frames were acquired at the rate of 1 frame every 4 s for 1 min 20 s and 15 min, respectively. FLIP was performed 30 days after the start of myelination.

## Ex vivo imaging of intercostal nerves in triangularis sterni explants

Triangularis sterni explants were isolated as previously described (*Kerschensteiner et al., 2008*; *Wang et al., 2021*) from mice expressing a SEP-Nfasc186 transgene on either a WT, Nfasc[-/-], or Cntnap1[-/-] background. Following dissection of the thorax, the nerve muscle explant was pinned using minutin pins (Fine Science Tools) in a 40 mm inner well of a 60 mm Sylgard 184-coated dish containing oxygenated (95% $O_2$, 5% $CO_2$) and cooled Ringers solution (125 mM NaCl, 2.5 mM KCl, 2 mM $CaCl_2$, 1 mM $MgCl_2$, 1.25 $NaH_2PO_4$, 26 mM $NaHCO_3$, 20 mM glucose). The dish was immediately placed on the stage of a Nikon Eclipse NI-E microscope in a temperature-controlled imaging chamber at 37°C (OkoLabs) and superfused (1 ml/min) with oxygenated Ringer's solution. A temperature probe (Warner Instruments) was placed in the dish close to the explant, which was kept at 37°C. The preparation was imaged for a maximum of 2.5 hr. The microscope was equipped with a x10/0.3 NA Plan Fluor long working distance objective lens and a Plan Achromat x100/1.10 NA water-immersion objective lens, Chroma ET filters, an Orca Flash 4.0v3 sCMOS camera (Hammamatsu Photonics), and LED illumination (CoolLED pE-4000) controlled by NIS-Elements AR software. FRAP was performed on nerve muscle explants from P1-P8 mice. An early cluster was identified as having flanking membrane SEP-Nfasc186 fluorescence (P1-P2) and the absence of myelin by phase contrast microscopy. A heminode (P3-P6) was identified when the adjacent heminode was in close proximity, and the identification of nodes (P6-P8) was confirmed by visualising flanking myelin by phase contrast microscopy. FRAP was performed using a focussed 405 nm laser (Rapp-OptoElectronic) (10% power for 17 cycles) for bleaching through the x100 objective on a focussed rectangle positioned on an axon using SysCon software to trigger a TTL (NI-DAQ) pulse triggering the laser and the UGA-40 Firefly scanner (Rapp OptoElectronic). Both DAPI and GFP Chroma filters were used for FRAP. Fluorescence recovery was recorded by time-lapse acquisitions every 4 s for 40 s (pre-bleach) and 15 min (post-bleach) using an exposure time of 100 ms. Camera pixel binning was 2x2. All images were corrected for drift using the Manual Drift Correction Plugin for ImageJ.

## Vesicle trafficking

For studies of vesicle trafficking in β1Nav-EGFP[+] DRG neuron-Schwann cell co-cultures, images were recorded every 500 ms for 1 min with a X100 (NA 1.46) objective. Vesicle movement was recorded

on the same set of axons immediately before nocodazole treatment and at the end of the 4 hr treatment. To control for potential FRAP/FLIP-induced phototoxicity, vesicle movement was recorded on the same set of β1Nav-EGFP$^+$ axons immediately before and immediately after performing FRAP/FLIP as described above. Vesicle movement was analysed using kymographs generated by KymoToolBox (an ImageJ plugin; *Zala et al., 2013*). The kymographs were manually traced to obtain vesicle speeds.

## Cluster counts

SEP-Nfasc186$^+$ or β1Nav-EGFP$^+$ early clusters, heminodes, and nodes were manually counted at the indicated time points on live DRG neuron-Schwann cell co-cultures. Clusters were categorised based on the criteria reported in the 'Materials and methods' section under 'Live cell Imaging of DRG cultures'. For each culture, counts were performed on the same area of the microfluidic device (measuring approximately 10 mm$^2$) at each time point. To obtain the total number of neurons present in the area, a tiled image of the area was acquired using a X10 objective and phase contrast illumination. DRG somas, which are easily recognizable by their round shape and large size, were then counted manually. The total number of SEP-Nfasc186$^+$ or β1Nav-EGFP$^+$ early clusters, heminodes, and nodes in the target area was then divided by the total number of DRG somas present in the same area, and the resulting figure was multiplied by 100 to obtain the number of clusters/100 neurons. To quantitate the number of early clusters at P1 in vivo in sciatic nerve sections from WT, β1Nav-EGFP, and SEP-Nfasc186 mice, we counted βIV-spectrin-positive clusters that lacked paranodal Caspr and adjacent myelin as a percentage of all nodal clusters in a minimum of five fields of view per mouse and three mice per genotype (two WT mice at P3) using a X40 objective and a Zeiss Apotome microscope. An early cluster is a nodal cluster without an adjacent Caspr-positive paranode or a P0-positive Schwann cell sheath, a heminode is a nodal cluster adjacent to a single Caspr-positive paranodal axo-glial junction that is adjacent to a P0-positive myelin sheath, and a node is a nodal cluster with two Caspr-positive paranodes. Representative images are shown using a X63 objective.

## Correlative light and electron microscopy

For CLEM, SEP-Nfasc186$^+$ DRG neuron-Schwann cell co-cultures were grown in microfluidic devices that had been transiently bonded to glass coverslips and imaged live 10 days after the induction of myelination. For each microfluidic device, up to three ROIs centred on representative examples of early clusters were selected for imaging. To facilitate re-identification of the selected ROIs after staining and resin embedding, the following set of dual-channel (GFP and bright-field) live images was acquired for each ROI: a 2x2 tiled z-stack with a X100 objective, a 2x2 tiled z-stack with a X40 objective, and a 6x6 tiled single-plane image with a X40 objective. Immediately after live imaging, co-cultures were fixed for 1 hr at RT in 2.5% glutaraldehyde (Agar Scientific), 4% formaldehyde in 0.1 M sodium cacodylate (Agar Scientific) buffer, pH 7.3, and then washed three times for 5 min in 0.1 M sodium cacodylate buffer. Co-cultures were then stained for 30 min at RT with a 1:300 dilution of BrainStain Fluorescent dye mix (ThermoFisher) in 0.1 M sodium cacodylate and washed three times for 3 min in 0.1 M sodium cacodylate. After staining, a second set of multichannel images of the target ROIs was acquired as described above. Samples were then stained with 0.1% tannic acid (Electron Microscopy Sciences) in 0.1 M sodium cacodylate buffer for 20 min and washed three times for 3 min in 0.1 M sodium cacodylate. Samples were then incubated in reduced osmium tetroxide (2% aqueous OsO$_4$, 1.5% potassium ferrocyanide in 0.1 M sodium cacodylate buffer) for 1 hr at RT. Samples were washed five times for 3 min in dH$_2$O, before incubation in 1% thiocarbohydrazide for 20 min at RT. After 5x3 min washes in dH$_2$O, samples underwent a second osmium staining (2% aqueous OsO$_4$) for 40 min. Samples were washed five times for 3 min in dH$_2$O. The PDMS device was gently peeled off the cover glass, taking care not to disrupt the underlying cell layer. Samples were then treated with lead aspartate (0.02 M lead nitrate, 0.03 M aspartic acid in dH$_2$O, pH 5.5) for 30 min at 60°C (*Walton, 1979*). Samples were dehydrated through a graded series of ethanol (50%, 70%, 90%), followed by two washes in 100% ethanol, and infiltrated with Agar 100 Premix resin 'hard' (AgarScientific), using ethanol: resin mixes at ratios 1:1, 1:2, and 1:3 for 30 min each, followed by two changes of 100% resin for 30 min. Samples were then covered in 100% resin and cured for 48 hr at 60°C. The cured resin was carefully separated from the glass. Excess resin was removed using a junior hacksaw and scalpel before the block was mounted onto a cryo pin, cell side up, using

a conductive silver epoxy compound. Targeted trimming was performed using an ultra-microtome (Leica) as previously described (*Booth et al., 2019*). Samples were painted with Electrodag silver paint (avoiding the block face) and then coated with 10 nm AuPd using a Q150T sputter coater (Quorum Technologies). The sample was inserted into the Gatan 3View sample holder and adjusted so that the block face would be central in the microtome and parallel with the knife edge. After loading into the Gatan 3View microtome, which was mounted in a Quanta FEG250 ESEM (FEI), the sample height was raised manually using the dissecting microscope until the block face was close to the height of the knife. The final approach of the block face to the knife was achieved using the automatic approach on Digital Micrograph (Gatan) at 200 nm. Progressive low-magnification 'survey' images (at ×600 magnification; 200 nm sections) were first acquired from the block with continued reference to optical images. Once a suitable landmark or ROI had been identified, appropriate section thickness and acquisition settings were established (kV 2.3, image size, 3180×6954; dwell time, 15μs; magnification ×1972; final pixel size = survey images - 35.9 nm, stacks containing early clusters – 5.4 nm and 4.9 nm; section thickness, 80 nm).

EM datasets were batch-converted into tiff files in preparation for modelling in Amira (Thermo-Fisher). Volume data were imported into Amira and the appropriate voxel dimensions inputted when prompted. CLEM registration was performed by scaled alignment of clearly visible biological fiducials (see *Figure 2—figure supplement 2*). Segmentation of LM (GFP) and EM volume data was performed using the blow tool.

## Quantification and statistical analysis

FIJI was used to view and analyse images and videos. For FRAP and FRAP-FLIP analysis of data collected from SEP-Nfasc186[+] DRG neuron-Schwann cell co-cultures and ex vivo triangularis sterni preparations, the mean fluorescence intensity of the bleached region was background-subtracted and corrected for acquisition photobleaching by multiplying it by a photobleaching correction factor. For each time point, the photobleaching correction factor was calculated by dividing the background-subtracted mean fluorescence intensity of a reference region (i.e., an independent axon in the same imaging field of view) at that time point by the background-subtracted mean fluorescence intensity of the same region at the first pre-bleach time point. Photobleaching-corrected intensities in the FRAP ROI were then normalized to the average pre-bleach fluorescence intensity of the same region, re-scaled so that the intensity of the first post-bleach frame was equal to zero, and plotted. The recovery fraction after photobleaching was calculated as the average of the last 10 data points in the FRAP recovery plot. For FLIP analysis, mean fluorescence intensities were measured at the node that was flanked by the two FLIP ROIs. Data were processed as above, but were not re-scaled before plotting. Fluorescence loss at the node was calculated as the average of the last 10 data points in the FLIP intensities plot. For the analysis of FRAP data collected from β1Nav-EGFP[+] DRG neuron-Schwann cell co-cultures, a control ROI was positioned on the target axon, adjacent to the FRAP ROI. For each time point, mean fluorescence intensities in the FRAP ROI and control ROI were background-subtracted, and averaged to obtain mean pre-bleach intensities (frames 1–20) and mean post-recovery intensities (frames 236–245, corresponding to the last 10 frames of the 15-min post-bleach acquisition series). When a single post-recovery frame was acquired 4 hr post-bleach, background-subtracted FRAP and control ROI mean fluorescence intensities from this single image were used instead. FRAP/control ROI ratios were then calculated from mean pre-bleach, post-FRAP, and mean post recovery fluorescence intensities. For the analysis of early cluster movements in SEP-Nfasc186[+] DRG neuron-Schwann cell co-cultures, the xy coordinates of individual clusters were recorded at each time point using the imageJ plugin MTrackJ (*Meijering et al., 2012*) and drift corrected using the xy coordinates of an immobile feature of the microfluidic device. The total displacement of each cluster was then calculated as the sum of the Cartesian distances between its planar coordinates between consecutive time points. All data are represented as mean ± SEM unless otherwise mentioned in the figure legends. Statistical analyses were performed using GraphPad Prism version 8.4.0 software. Statistical significance was analysed by two-tailed Student's t-test or ANOVA followed by the multiple comparisons test indicated in the figure legends. Sample sizes are reported in the corresponding figure legends. The sample size was determined based on our previous work (*Ghosh et al., 2020*). A p-value<0.05 was considered to be statistically significant.

## Acknowledgements

We thank Qiushi Li and Andrew Garrie for their invaluable technical support. Alison Beckett and Ian Prior (both Liverpool University) are thanked for help and advice with CLEM. Manzoor Bhat is thanked for the generous gift of an antibody against Caspr. This work was supported by a grant from the Wellcome Trust to PJB (grant no. 107008). PJB is a Wellcome Trust Investigator. DGB is funded by a Nottingham University Research Fellowship.

## Additional information

### Funding

| Funder | Grant reference number | Author |
|---|---|---|
| Wellcome Trust | 107008 | Peter J Brophy |
| Wellcome Trust Investigator | | Peter J Brophy |
| Nottingham University Research Fellowship | | Daniel G Booth |

The funders had no role in study design, data collection and interpretation, or the decision to submit the work for publication.

### Author contributions

Elise LV Malavasi, Conceptualization, Data curation, Formal analysis, Validation, Investigation, Methodology, Writing - original draft, Writing - review and editing; Aniket Ghosh, Formal analysis, Investigation, Methodology, Writing - review and editing; Daniel G Booth, Formal analysis, Investigation, Methodology; Michele Zagnoni, Resources, Methodology; Diane L Sherman, Investigation, Methodology, Writing - review and editing; Peter J Brophy, Conceptualization, Supervision, Funding acquisition, Writing - original draft, Project administration, Writing - review and editing

### Author ORCIDs

Elise LV Malavasi (iD) https://orcid.org/0000-0002-2240-0553
Aniket Ghosh (iD) http://orcid.org/0000-0002-3771-6390
Diane L Sherman (iD) http://orcid.org/0000-0002-3104-6656
Peter J Brophy (iD) https://orcid.org/0000-0002-0262-9545

### Ethics

Animal experimentation: All animal work conformed to UK legislation (Scientific Procedures) Act 1986 and to the University of Edinburgh Ethical Review policy and was performed under Project Licence No. P0F4A25E9 from the Animals in Science Regulation Unit of the UK Home Office.

### Decision letter and Author response

Decision letter https://doi.org/10.7554/eLife.68089.sa1
Author response https://doi.org/10.7554/eLife.68089.sa2

## Additional files

### Supplementary files

- Source data 1. Early clusters in vivo.
- Source data 2. Early clusters in *Nfasc*$^{-/-}$ neurons.
- Source data 3. Early clusters in *Cntnap1*$^{-/-}$ neurons.
- Source data 4. Blebbistatin treatment.
- Source data 5. Percentage and speed of mobile early clusters.
- Transparent reporting form

### Data availability

All source data files have been provided.

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
