## [Decision Letter]

**Acceptance summary:**

The manuscript makes the important finding that clusters of proteins appear at a very early stage of formation of nodes of Ranvier. Based upon an extensive analysis, the Malavasi et al. study presents evidence that the earliest clusters precede heminode formation. The analysis suggests the process and progression of early clusters represents a first step in the sequential assembly of the nodes of Ranvier. An important question, as saltatory conduction depends upon the formation and trafficking of nodal proteins.

**Decision letter after peer review:**

Thank you for submitting your article "Dynamic early clusters of nodal proteins contribute to node of Ranvier assembly in the peripheral nervous system" for consideration by *eLife*. Your article has been reviewed by 3 peer reviewers, and the evaluation has been overseen by a Reviewing Editor and Kenton Swartz as the Senior Editor. The following individual involved in review of your submission has agreed to reveal their identity: Fabrice Ango (Reviewer #2).

Summary:

This work aims to further dissect the earliest events in node of Ranvier formation in the peripheral nervous system. This is an important question since saltatory conduction is essential for efficient nervous system function. The methods are appropriate and support the major claims of the manuscript.

Essential Revisions:

The manuscript was evaluated by three reviewers who felt the study is of considerable interest and that the methods are appropriate and give interpretable results. However a number of important issues will require essential revisions.

1) A major point is the lack of in vivo evidence for the early clusters. Since the experiments used transgenic mice, overexpression may provide artifactual results. Evidence of the appearance of these clusters in vivo should be provided using the transgenes, as well as in plain wild type mice. In this regard, it is not clear whether Figure 1S2A is from a transgenic DRG co-culture; Figure 1S2B is from WT mice.

2) Additional quantitation of the early clusters need to be provided, as described below.

3) The role of myelination in the formation of protein clusters should be clarified. The involvement of gliomedin in cluster formation should also be explained. The reviews indicated that several statements and phrases about cluster formation need to be qualified.*Reviewer #1:*

This paper reports the existence of clusters of proteins, normally associated with nodes of Ranvier, along axons before they become myelinated. The question of whether these Na^+^ channel-containing protein complexes require myelination has been the subject of many studies and seems to still be controversial. The authors conclude that the 'early clusters' constitute the earliest stage of node of Ranvier formation. There are many things to like about this paper – the strongest results include careful and interesting studies of the dynamic nature of the proteins associated with the 'early clusters', heminodes, and nodes. State of the art tools and techniques are used to investigate these dynamic structures. However, the major limitation of the study is that almost all experiments were performed in vitro and one of the main conceptual findings about the existence of 'early clusters' can be attributed to a culture artifact – namely the presence of excess gliomedin. Their data and previous publications do not support the existence of 'early clusters' as a prominent structure in the peripheral nervous system in vivo.

1. There is little evidence provided that these early clusters form in vivo during development. The authors show a single example in vivo (Figure 1S2B) and there is no quantification at all. Old studies like Lambert et al. 1997 – which is referred to as evidence to support their claim – should be revisited experimentally with much improved antibodies and microscopy. The authors should report how frequently they observe these clusters in vivo.

2. The reported in vitro results can be entirely explained by the presence of secreted gliomedin in the cultures. The phenomenon of clusters along axons not associated with myelin was reported in Eshed-Eisenbach et al. 2020. However, these clusters were never detected in the absence of gliomedin (figure 2A from their paper). Besides looking at the gliomedin KO mice, one more simple way to show the relationship would be to use their compartment model and induce myelination in the axonal compartment rather than the somal compartment. They could then test the idea they state on line 578-580. This reviewer predicts the early clusters will be present in the axonal compartment before the somal compartment, indicating they are driven by the induction of myelination and are Schwann cell dependent.

*Reviewer #2:*

In their manuscript, Malavasi et al. investigated how nodal proteins contribute to the node of Ranvier assembly in the peripheral nervous system. They used live imaging on an in vitro microfluidic co-culture model of DRG and Schwann cells. They found that the formation of early nodal proteins cluster before heminode assembly is required for normal development and formation of myelinated axons. Interestingly, these early clusters of highly mobile nodal protein complexes diffuse along the axonal membrane independently of heminodes or axo-glial junctions. The fusion of these early clusters contributes to the formation of heminodes and nodes.

They confirmed their data using correlative electron microscopy and several mutant mice. Although the mechanism responsible for the mobility of these early clusters is not completely understood, they appear as the earliest clustering mechanism that leads the assembly of nodes of Ranvier. All experiments are well thought, and the analysis is rigorous.

– In this co-culture model, do the authors identify early clusters of Neurofascin186 without expression of the transgenic form proteins?

– Since the authors suggest that specialized glial interaction with the axon is not necessary for the formation of early clusters, I was wondering whether it will be possible to use a conditioned medium obtained from the co-culture DRG-Schawn cells to induce early cluster formation in DRG neurons in myelinating conditions (to complete Figure 2A).

– In their final model (Figure 7c), I understand that SC (dark grey; left) will later stabilize heminode formation. I don't understand what is the structure in light grey (right) around the axon?*Reviewer #3:*

Development of the Nodes of Ranvier in the PNS involves two compensating mechanisms, the clustering of Nfasc and voltage-gated sodium channels (Nav) by gliomedin produced by Schwann cells, and the action of the paranodal axo-glial junctions that prevent the lateral diffusion of nodal proteins to the internodes. Malavasi et al. describes the formation of early nodal like clusters along axons at sites that contact Schwann cells prior to the formation of microvilli, which are thought to represent the earliest location of nodal protein clustering. They use DRG neurons isolated from transgenic mice expressing fluorescently-tagged Nfasc186 or b1Nav proteins to show that these clusters are highly mobile, fuse to each other, incorporate into heminodes, or disintegrate. The formation of these clusters depends on Nfasc but not on caspr. They also show that the lateral movement of early clusters is impaired after disrupting actin polymerization, although it is not clear whether the latter is required in neurons or in Schwann cells. Based on these findings, the authors conclude that the formation of early clusters consist of the first step in the assembly of the nodes of Ranvier. The current work extends the observation of Eshed-Eisenbach et al., (2020), who reported the formation of ectopic nodal like clusters in wild type myelinating cultures, as it shows the fate of these clusters.

There are several points of concern that should be addressed:

1. The work is based on the use of myelinating cultures, previously shown to contain gliomedin which induces the formation of ectopic clusters. Whether this is also part of the mechanism of node assembly in vivo requires a quantitative developmental analysis of nodal components in both transgenes. Is the cluster shown in figure S2 exist below MAG-labeled internode? How frequent are these early clusters observed compared to heminodes in the developing sciatic nerve?

2. The time laps imaging, as well as their derived still images (i.e., Figure 5 and 7) should include myelin markers such as MAG, MBP and P0 in order to clarify what we are looking at.

3. Page 8, figures 2A and figure 6A, please clarify how "cluster/100 neurons" was determined.

4. The authors make several claims that may not be justified. For example, that the work describe "a third mechanism that can contribute to PNS node formation" (page 40), or that the work "expand the repertoire of mechanistic redundancy that has characterized the evolution of saltatory conduction in vertebrate PNS" (page 38). The mechanism involves the clustering of Nfasc186 likely by gliomedin expressed by Schwann cells, which as they show, are found in close proximity with the axon. This is one of the known mechanisms that was previously thought to function mainly at heminodes, hence does not really represent a new redundant mechanism. Similarly, to claim that the "data reveal that early cluster assembly is spatially and temporally regulated developmental stage of PNS myelination" (page 33) requires in vivo analysis which is currently lacking.

[Editors' note: further revisions were suggested prior to acceptance, as described below.]

Thank you for resubmitting your work entitled "Dynamic early clusters of nodal proteins contribute to node of Ranvier assembly during myelination of peripheral neurons" for further consideration by *eLife*. Your revised article has been evaluated by Kenton Swartz (Senior Editor) and a Reviewing Editor.

The editors appreciate your response to the reviews of your manuscript. The paper makes the proposal that clusters of proteins appear at a very early stage in the formation of nodes of Ranvier. While the majority of comments have been properly addressed, a reservation remains about the in vivo evidence for early clusters, which had been requested by the previous reviewers. The results from the Eshed-Eisenbach Neuron paper in 2020 is cited as published evidence of in vivo clustering of Na^+^ channels.

The data from the Eshed-Eisenbach paper showed that clusters were associated with gliomedin, which is a principal clustering mechanism. Although the work of Eshed-Eisenbach demonstrated a dependency on gliomedin and not paranodal junctions, the study did not distinguish whether early clusters of nodal proteins preceded heminodes and are non-heminodal in nature. Based upon in vitro analysis, the revised Malavasi et al. manuscript argues that the earliest clusters precede heminode formation and represents a new mechanism to explain the process of node formation. Although a subtle point, the Eshed-Eisenbach study did not conclusively establish the early clusters are non-heminodal. Hence in vivo evidence is still needed to support the existence of dynamic early clusters in the absence of gliomedin.

In summary, more evidence is needed to show that non-heminodal clusters occur in vivo to support the claims in the revised manuscript. Prior reviewers suggested that this would require a developmental analysis to examine MBP or P0 staining to reveal the nascent myelin sheath and the existence of early clusters without any associated flanking myelinating Schwann cell. The editors recommend that this issue be addressed experimentally. Otherwise the proposed alternative model will need to be qualified.

---

## [Author Response]

1) A major point is the lack of in vivo evidence for the early clusters. Since the experiments used transgenic mice, overexpression may provide artifactual results. Evidence of the appearance of these clusters in vivo should be provided using the transgenes, as well as in plain wild type mice. In this regard, it is not clear whether Figure 1S2A is from a transgenic DRG co-culture; Figure 1S2B is from WT mice.

The existence of early clusters in vivo in the developing peripheral nerves of mice is not in doubt and has been quantitated robustly by Eshed-Eisenbach et al., (2020) Neuron, 106, 806-815, as documented in their Figures 4C and 4D. These authors used mice doubly homozygous for floxed *Bmp1* and *Tll1* alleles lacking a Cre-driver as their controls, which they described as WT. The absence of a peripheral nerve phenotype in Eshed-Eisenbach et al., showed that these controls are effective surrogates for the study of cluster formation in WT mice. The same doubly floxed mice were generated and characterised previously by Muir et al., (2014) Hum.Mol.Gen. who showed that insertion of LoxP sites did not disrupt gene expression, hence, they described them as phenotypically WT.

The developmental relationship between these “single clusters” and nodal clusters flanked by paranodal Caspr observed in vivo in Eshed-Eisenbach et al., was consistent with what we observed in the coculture system (our Figure 6): that is, early clusters appear first, and then decline in number in favour of heminodes and nodes.

We now draw more attention to the relevant data of Eshed-Eisenbach et al., in the Introduction (lines 40-43) and invite parallels with our own data in Figure 6 in the Discussion (lines 438-441).

Figure 1 – Supplement 2 now includes images of axons in WT cocultures (Figure 1–Supplement 2A), and in WT sciatic nerves in vivo (Figures 1–Supplement 2B and 1–Supplement 2C) and β1Nav-EGFP and SEP-Nfasc186 mice in vivo. (legend Figure 1-Supplement 2). This is now emphasized in the Results (104-113).

2) Additional quantitation of the early clusters need to be provided, as described below.

Notwithstanding the evidence from both previous work and this study concerning the existence of early clusters in developing peripheral axons in vivo summarised above, images of early clusters in sciatic nerves in WT, β1Nav-EGFP and SEP-Nfasc186 mice in vivo at P1 (P1 is the day of birth) are now included (Figure 1-Supplement 2C), and quantitated (lines 104-113, and 774-782). Antibodies used for quantitation are added to the Key Resources Table and highlighted in yellow.

3) The role of myelination in the formation of protein clusters should be clarified. The involvement of gliomedin in cluster formation should also be explained. The reviews indicated that several statements and phrases about cluster formation need to be qualified.

The role of myelination in cluster formation is directly addressed in our Figure 2A where we show that early clusters in DRG neuron axons require both the presence of Schwann cells and myelinating conditions. The involvement of gliomedin in early cluster formation, as established by Eshed-Eisenbach et al., (2020), is not in doubt, and our data provide evidence that it is the interaction of gliomedin with neuronal Neurofascin 186 that mediates this effect. This point is addressed further in response to Reviewer #1, point 2.

Reviewer #1:1. There is little evidence provided that these early clusters form in vivo during development. The authors show a single example in vivo (Figure 1S2B) and there is no quantification at all. Old studies like Lambert et al. 1997 – which is referred to as evidence to support their claim – should be revisited experimentally with much improved antibodies and microscopy. The authors should report how frequently they observe these clusters in vivo.

This is answered in responses to Essential Revisions, points 1 and 2. The thorough piece of work reported by Eshed-Eisenbach et al., (2020) is also referenced more clearly.

2. The reported in vitro results can be entirely explained by the presence of secreted gliomedin in the cultures. The phenomenon of clusters along axons not associated with myelin was reported in Eshed-Eisenbach et al. 2020. However, these clusters were never detected in the absence of gliomedin (figure 2A from their paper). Besides looking at the gliomedin KO mice, one more simple way to show the relationship would be to use their compartment model and induce myelination in the axonal compartment rather than the somal compartment. They could then test the idea they state on line 578-580. This reviewer predicts the early clusters will be present in the axonal compartment before the somal compartment, indicating they are driven by the induction of myelination and are Schwann cell dependent.

We have confirmed that, as previously reported by Eshed-Eisenbach et al., 2020, gliomedin is enriched at early clusters (Figure 1-Supplement 1). The requirement for gliomedin for the formation of early clusters (Eshed-Eisenbach et al., 2020) through its interaction with axonal Neurofascin (our work) is clear.

The data presented in Figure 6 confirms the Reviewer’s prediction. Namely, that under identical culture conditions, early clusters (and heminodes and nodes) assemble in the proximal axonal segments (located in the somal compartment) before appearing in more distal axonal segments (located in the axonal compartment) and that assembly of early clusters in vitro is not only Schwann cell dependent, but also requires myelinating conditions (Figure 2A). The presence of Schwann cells is insufficient for cluster formation, they must also be myelinating (lines 144-145).

The title of the legend has been changed to emphasise this point (legend Figure 2).

Additional, possibly axon-driven mechanisms, may differentially regulate the way axons respond to gliomedin exposure and its interaction with axonal Neurofascin in proximal vs distal axonal segments, and it will be of interest to address this in the future.

Reviewer #2:– In this co-culture model, do the authors identify early clusters of Neurofascin186 without expression of the transgenic form proteins?

Yes, see Figure 1-Supplement 2A.

– Since the authors suggest that specialized glial interaction with the axon is not necessary for the formation of early clusters, I was wondering whether it will be possible to use a conditioned medium obtained from the co-culture DRG-Schawn cells to induce early cluster formation in DRG neurons in myelinating conditions (to complete Figure 2A).

This issue has been addressed previously (Eshed-Eisenbach 2005, Neuron) who confirmed that Nav clusters form on isolated DRG neurons in the absence of Schwann cells when DRG neurons are exposed to a recombinant gliomedin fragment that specifically binds to Nfasc186.

– In their final model (Figure 7c), I understand that SC (dark grey; left) will later stabilize heminode formation. I don't understand what is the structure in light grey (right) around the axon?

This represents a Schwann cell that has not started ensheathing the axon. We know that these are in close proximity to the axon where early clusters are present based on our staining and CLEM data (Figure 2). We have now amended the legend to Figure 7 to include a fuller explanation of the different structures/colours **(**legend to Figure 7). We hope this is now clearer.

Reviewer #3:There are several points of concern that should be addressed:1. The work is based on the use of myelinating cultures, previously shown to contain gliomedin which induces the formation of ectopic clusters. Whether this is also part of the mechanism of node assembly in vivo requires a quantitative developmental analysis of nodal components in both transgenes. Is the cluster shown in figure S2 exist below MAG-labeled internode? How frequent are these early clusters observed compared to heminodes in the developing sciatic nerve?

Quantitation of the development of early clusters in comparison with nodal clusters associated with paranodes in vivo was reported in Figure 4 of Eshed-Eisenbach et al., 2020 and we have repeated quantitation of early clusters as detailed in answers to Essential Revisions, points 1 and 2. The time course of early cluster appearance with respect to heminodes in culture is shown in our Figure 6 and is in broad agreement with the data of Eshed-Eisenbach et al., 2020.

2. The time laps imaging, as well as their derived still images (i.e., Figure 5 and 7) should include myelin markers such as MAG, MBP and P0 in order to clarify what we are looking at.

Where individual axonal segments were not stained for myelin markers after time-lapse imaging, live bright field images of the same axonal segments were included with the still live fluorescence images to demonstrate the absence or presence of myelin. These are readily distinguishable in bright field and a dotted magenta line traces the outline of these myelin sheaths, for example in Figures 2-Supplement 1, 3, 4-Supplement 2, 5-Supplement 1, and 7.

3. Page 8, figures 2A and figure 6A, please clarify how "cluster/100 neurons" was determined.

This is explained in the “Cluster counts” paragraph in the Methods section. This section has been amended to make it clearer (lines 774-782).

4. The authors make several claims that may not be justified. For example, that the work describe "a third mechanism that can contribute to PNS node formation" (page 40), or that the work "expand the repertoire of mechanistic redundancy that has characterized the evolution of saltatory conduction in vertebrate PNS" (page 38). The mechanism involves the clustering of Nfasc186 likely by gliomedin expressed by Schwann cells, which as they show, are found in close proximity with the axon. This is one of the known mechanisms that was previously thought to function mainly at heminodes, hence does not really represent a new redundant mechanism. Similarly, to claim that the "data reveal that early cluster assembly is spatially and temporally regulated developmental stage of PNS myelination" (page 33) requires in vivo analysis which is currently lacking.

By “a third mechanism that can contribute to PNS node formation” we are not referring to the clustering of Nfasc186 at early clusters by gliomedin. This is a well-established phenomenon. By novel redundant mechanism we are referring to the active migration of early clusters towards and fusion with nascent PNS heminodes. This has not been reported previously. The text is amended to make this point clearer (lines 438-441).

[Editors' note: further revisions were suggested prior to acceptance, as described below.]

The data from the Eshed-Eisenbach paper showed that clusters were associated with gliomedin, which is a principal clustering mechanism. Although the work of Eshed-Eisenbach demonstrated a dependency on gliomedin and not paranodal junctions, the study did not distinguish whether early clusters of nodal proteins preceded heminodes and are non-heminodal in nature. Based upon in vitro analysis, the revised Malavasi et al. manuscript argues that the earliest clusters precede heminode formation and represents a new mechanism to explain the process of node formation. Although a subtle point, the Eshed-Eisenbach study did not conclusively establish the early clusters are non-heminodal. Hence in vivo evidence is still needed to support the existence of dynamic early clusters in the absence of gliomedin.In summary, more evidence is needed to show that non-heminodal clusters occur in vivo to support the claims in the revised manuscript. Prior reviewers suggested that this would require a developmental analysis to examine MBP or P0 staining to reveal the nascent myelin sheath and the existence of early clusters without any associated flanking myelinating Schwann cell. The editors recommend that this issue be addressed experimentally. Otherwise the proposed alternative model will need to be qualified.

As requested, we have now quantitated early clusters (clusters lacking both adjacent paranodal Caspr and P0-positive myelin), heminodes (clusters with a single Caspr-positive paranodal axoglial junction adjacent to a P0-positive myelin sheath) and nodes (clusters flanked by Caspr-positive paranodal axoglial junctions) at three ages, P1, P3 and P5 in WT sciatic nerves and the criteria used to classify these nodal clusters in vivo are now explicit in the Methods.

These data are shown in Figure 6B and 6C and the source data has been added to the source data files. As requested, these new data show more evidence that early (non-heminodal) clusters occur in vivo and that they precede the appearance of heminodes, which, in turn, precede nodes.

We thank the reviewers for their comments which have helped to improve the manuscript. We trust that we have addressed the requested revisions adequately and look forward to your reply.